# Reaction-diffusion in a growing 3D domain of skin scales generates a discrete cellular automaton

Anamarija Fofonjka[1,2] & Michel C. Milinkovitch [1,2✉]

We previously showed that the adult ocellated lizard skin colour pattern is effectively generated by a stochastic cellular automaton (CA) of skin scales. We additionally suggested that the canonical continuous 2D reaction-diffusion (RD) process of colour pattern development is transformed into this discrete CA by reduced diffusion coefficients at the borders of scales (justified by the corresponding thinning of the skin). Here, we use RD numerical simulations in 3D on realistic lizard skin geometries and demonstrate that skin thickness variation on its own is sufficient to cause scale-by-scale coloration and CA dynamics during RD patterning. In addition, we show that this phenomenon is robust to RD model variation. Finally, using dimensionality-reduction approaches on large networks of skin scales, we show that animal growth affects the scale-colour flipping dynamics by causing a substantial decrease of the relative length scale of the labyrinthine colour pattern of the lizard skin.

[1] Laboratory of Artificial & Natural Evolution (LANE), Dept. of Genetics & Evolution, University of Geneva, Geneva, Switzerland. [2] SIB Swiss Institute of Bioinformatics, Geneva, Switzerland. ✉email: Michel.Milinkovitch@unige.ch

In fish, amphibians and reptiles, dermal chromatophores[1–3] are pigmentary cells (melanophores, xanthophores and erythrophores that accumulate brown/black, yellow and red pigments, respectively) as well as iridophores that contain spatially-organised lattices of guanine nanocrystals[4] generating structural colours by light interference[5]. Mammals and birds do not exhibit iridophores but produce brown, black and reddish melanic pigments that their melanocytes transfer to hair or feathers[6]. In birds, quasi-periodic arrays of keratinised nanostructures and/or melanin granules additionally produce a large set of structural colours[7], whereas the skin colour of snakes and lizards, dominated by dermal chromatophores, can also be supplemented by structural (generally iridescent) colours due to 'nanogratings', i.e., submicron-sized periodic skin-surface deformations[8].

In many animals, these colours form a collection of shapes, such as stripes, spots, tessellations or meanders, that collectively make a symmetry-breaking skin pattern. These patterns form because chromatophores spatially segregate during development through short-range and long-range interactions[9–11]. Note that individuals typically exhibit a fixed pattern. Indeed, although movements of pigmentary intracellular compartments[12], and possibly active tuning of photonic crystal geometry within iridophores[4,13], can generate (within minutes) behavioural changes of skin brightness and hue (for camouflage, thermoregulation and display; ref. [14]), the geometry of the pattern itself generally remains unchanged. On the other hand, skin patterns can dramatically change at larger time scales, i.e., it takes a substantial time for the patterning process to reach steady state. In many amniote species, the pattern is established during embryonic development and remains fixed after birth. In others, however, the skin colour pattern dramatically changes after birth, i.e., they exhibit drastically-different juvenile and adult patterns. These species provide the opportunity to readily investigate the dynamics of skin colour pattern formation, mitigating the general inaccessibility of embryos (especially in amniotes) in other species whose skin colour pattern development is completed before birth.

Whereas the underlying microscopic dynamical system involves discrete biological cells[9–11], the process of macroscopic pattern development can be considered continuous and quantitatively investigated (using nonlinear partial differential equations; PDEs) as a canonical reaction-diffusion (RD) system[15,16], well-known to generate various patterns at specific spatial scales[17–20]. For example, the network of short- and long-range activations and inhibitions among chromatophores that controls the development of the zebrafish striped colour pattern can be translated into a system of PDEs that accurately reproduce zebrafish patterns of wild-type and mutant lines[21]. Note that, 'cells-as-agents' approaches[22–24] (referred to as 'agent-based' (AB) below), some of them[23] incorporating the likely important role of iridophores[25,26], have also been successfully applied to model colour pattern formation in zebrafish.

Yet, some (but not all) species of snakes and lizards exhibit skin colour patterns made of juxtaposed skin scales of different colours, whereas each individual skin scale is monochromatic. This observation seems to conflict with the autonomous self-organisational patterning process predicted both by RD and AB models. Recently, our investigation of the post-hatching development of such a 'pointillist' skin colour pattern in the ocellated lizard (*Timon lepidus*) generated the following three realisations[27].

First, we identified[27] that the juvenile pattern of brown skin with white spots gradually transforms, on the dorsal trunk of the animal, into an adult labyrinthine pattern made of contrasting black and green chains of scales (Fig. 1). More specifically, high-resolution dorsal skin 3D geometry and colour-texture reconstructions in individual lizards at multiple time-points across the 4 years of their pattern development identified that an initial

phase of transformation of all juvenile white, light-brown and dark-brown scales into green or black scales is followed by a continuous, but rate-decreasing, process of green to-black and black-to-green colour switching of individual scales. These dynamics gradually generate a spatial distribution of green and black scales far from randomness[27], indicating that the colour switching process is not scale autonomous.

Second, our analysis of these time series of scale-colour switching indicated that the lattice of skin scales in ocellated lizards is a probabilistic cellular automaton (CA; refs. [28–31]) that dynamically computes the adult labyrinthine pattern[27], i.e., the colour state of each scale iteratively changes as a function of the states of its neighbours. This result was the first evidence that von Neumann's cellular automata, long considered as entirely abstract computational systems, have actually been generated by biological evolution[27].

Third, we demonstrated that, in a system of continuous RD equations, reduction of all diffusion coefficients by a factor >80% along the scale boundaries produces the emergence of not only a scale-by-scale colour pattern, but also discrete CA dynamics of scale-colour switching: the colour of individual scales tends to remain either homogeneously green or homogeneously black but can rapidly switch between these two states without maintaining intermediate values[27]. During this process, each scale remains, at all times, essentially monochromatic, whereas abrupt spatial variations between green and black skin always take place at scale boundaries, despite that the system is controlled by continuous RD equations. These observations indicate that each of the three interacting morphogens effectively exhibits an homogeneous concentration within a scale (because of large diffusion rates), allowing us to formally derive, from Turing's continuous RD equations, a discrete RD model (with skin scales as discretisation units) by renormalising the diffusion term[27]. In other words, the skin 3D geometry would transform a Turing continuous RD system of microscopic nonlinear interactions among chromatophores into a discrete macroscopic von Neumann CA (at the spatial scale of skin scales) that computes a pattern.

As all our previous RD simulations[27] included only two spatial dimensions (2D), the entirety of our demonstration relied on the explicit assumption that variations in the third spatial dimension of the ocellated lizard's skin (i.e., the presence of thick skin scales with much thinner borders) justifies a substantial reduction (by a factor $1-P > 0.8$) of the continuous RD diffusion coefficients at the one-dimensional edges of 2D scales, effectively decoupling the RD dynamics inside versus between scales. Here, we test the validity of this key assumption by performing RD numerical simulations in a three-dimensional (3D) domain of skin scales such that we 'let the 3D geometry do the job', i.e., we dispense with the need to use an artificial diffusion-reduction factor at scale borders. Given their high computational intensity, these 3D numerical simulations are ported on GPUs.

Our results show that this scheme produces a scale-by-scale colouring and a CA behaviour, not only with 3D lattices of idealised hexagonal prisms, but also with realistic quasi-hexagonal lattices of 3D scales reconstructed from episcopic microscopy data. In other words, solution of the system of RD equations on the reconstructed actual 3D geometry of the skin scales produces a spatially-discrete pattern exactly superposed on the lattice of scales, validating the central assumption used in our previous study[27]. We also show here that the 'snapping' of the pattern to the borders of the scales is robust to variation of the RD model, i.e., it occurs with vastly different sets of PDEs.

To facilitate future exploration of patterning dynamics generated by various RD models and geometries in scaled skins, we then propose a dimensionality-reduction approach that allows performing the simulations in 2D while integrating skin thickness

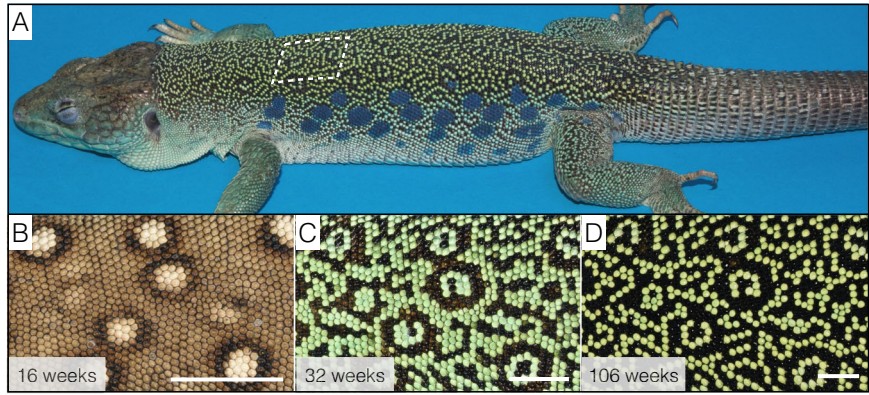

**Fig. 1 Colour pattern ontogeny in ocellated lizards. A** Adult ocellated lizards (here, a 3-year-old male) exhibit a labyrinthine dorsal pattern made of contrasting black and green chains of scales; we ignore here the lateral blue ocelli and the ventral scales. **B–D** Dorsal pattern time evolution from the juvenile brown skin with white ocelli to the nearly adult pattern. Note that remnants of the initial pattern organisation (mostly the position of white ocelli) are somewhat visible at the age of 106 weeks (**D**) but get obliterated in older lizards. Scale bars: 5 mm.

as position-dependent RD components' concentrations and effective diffusion coefficients. This scheme is slightly less accurate than bona fide 3D simulations but is more realistic than the reduction of diffusion coefficients along 1D edges of a 2D lattice of polygonal scales. Yet, we also show that the emergence of a CA behaviour is robust to alterations of the real skin geometry as all approaches generate very similar qualitative outcomes, i.e., non-trivial reduction of skin thickness at scale boundaries is sufficient for a transition between continuous and discrete patterns.

Finally, given that RD systems are known not to scale (i.e., the absolute length scale of the pattern should remain invariant), we explore the effect of growth on the skin colour patterning process in ocellated lizards. We show that the growth of the animal, across 3 years after hatching, deeply affects the dynamics of scale-colour change and produces a gradual decrease of the relative length scale of the pattern (measured in number of scale diameters) and a substantial increase of the average curvature of the pattern motifs. These results indicate that growth impacts form not only because of mechanical aspects of development[32,33] but also in the emergence of anatomies controlled by RD-like systems.

## Results

We use the interaction network and system of partial differential RD equations of Nakamasu et al.[21] (see Methods) that efficiently reproduce in 2D both zebrafish skin colour patterns[21] and the dynamics of ocellated lizard skin colour pattern development[27]. We extend this framework by implementing a 3D GPU-based finite-difference numerical approach[34] to test whether 3D geometry alone can explain the emergence, in a RD system, of a scale-by-scale coloration and CA dynamics of pattern development in the ocellated lizard. Our objective is to test whether these phenomena emerge in a 3D domain with periodic thickness variation when diffusion coefficients are maintained constant everywhere in the domain. In other words, we test if our hypothesis of applying a factor of diffusion reduction at scale borders of a 2D model[27] is entirely justified by the 3D geometry of the skin.

**Geometry transforms continuous RD into discretised systems.** First, we simulate the network of 3D scales as a regular lattice of hexagonal prisms (Fig. 2) of height $h$ connected by inter-scale domains of height $h_p$. Trivially, when $h_p/h = 1$ (Fig. 2A), there is no variation of thickness in the simulated 3D skin such that, at steady state, the Turing pattern is continuous and identical in top and bottom views (Fig. 2B) and coloration does not co-vary with

the positions of scale borders (Fig. 2C). On the other hand, when $h_p/h$ is reduced (Fig. 2D), the steady-state pattern becomes discrete (Supplementary Fig. 1). For example, when $h_p/h = 0.3$ (i.e., the approximate value observed in real scales), the concentration-gradient maxima of all RD components sharply co-localise with the positions of scale borders such that each scale, in top view, is either homogeneously green or homogeneously black (Fig. 2E, left panel; Fig. 2F). Note that, because inter-scale diffusion is unconstrained in the most basal part of the simulation domain, the concentration-gradients maxima less sharply coincide with the scale borders when the simulated skin is observed in bottom view (Fig. 2E, right panel). However, because the inter-scale skin thickness ($h_p$) is much smaller than the pattern length scale, the patterns in bottom and top views are highly similar, i.e., the effect generated by the separation of prisms in the upper part of the simulation domain propagates essentially all the way down to the base of the skin (Fig. 2F, bottom panel).

Second, we test the potential generality of the causal relationship between 3D geometry and the continuous-to-discrete transition of RD systems. In other words, we analyse whether thickness periodic variation of the 3D domain is sufficient for a RD system to generate a scale-by-scale pattern, irrespective of the specific system of nonlinear PDEs that are used. Figure 2G–J shows that both the Grey-Scott[35,36] and the Schnakenberg[37] two-component RD models (see Methods) generate a discrete pattern with transitions in components' concentrations that sharply coincide with the positions of hexagonal 3D prism borders. Note that the same phenomenon of scale-by-scale coloration is generated with alternative initial conditions (Supplementary Fig. 2). We therefore conclude that the effect of 3D geometry on pattern formation is likely to be very general, i.e., robust to variations of the RD model. Given that skin thickness variation affects the resulting pattern, the length scale of the steady-state patterns in a domain of constant thickness can easily be derived, for all these RD systems (three-component Nakamasu et al.[21], as well as two-component Grey-Scott[35,36] and Schnakenberg[37] models), from the simulations without scale boundaries (Fig. 2C and right panels in Fig. 2G, I) and compared to the size of the hexagonal elements.

**Actual skin geometry generates a discretised pattern in ocellated lizards.** Third, because one could argue that the discretisation of the pattern obtained with hexagonal prisms (Fig. 2D) is artificially enforced by the sharp transitions among neighbouring prisms (Fig. 2D), we simulate the same three-component RD model in a domain consisting in a quasi-hexagonal lattice of 3D

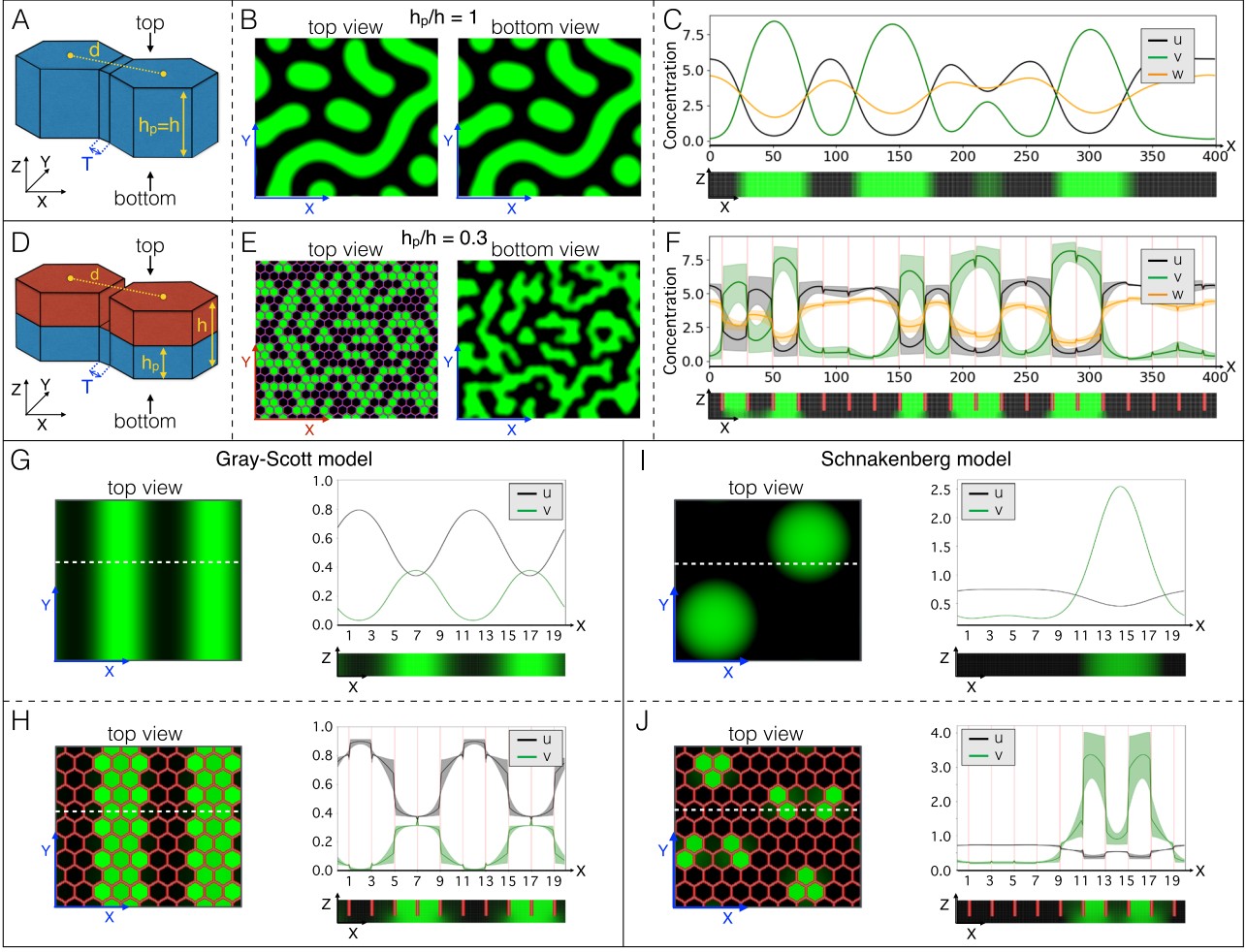

**Fig. 2 Geometry transforms continuous RD into discretised systems. A–C** When the domain does not exhibit variation of thickness (**A**), the steady-state RD pattern does not vary between top and bottom views of the simulated skin (**B**), and the concentrations of RD components show continuous spatial variation (**C**; concentration profile at y=0 averaged across z) and no variation in the z direction (bottom panel in **C**). **D–F** When the domain thickness is substantially smaller (hp) in between (T) than within (h) prisms (**D**; hp/h = 0.3), a prism-by-prism coloration emerges (**E**) with sharp transitions of concentrations of RD components at prisms' borders (**F**; average concentrations at y = 0; shaded regions are minimum-maximum concentration intervals in the z direction). Broader gradients at borders of prisms in bottom view (see also bottom panel of **F**) are due to unconstrained diffusion in the basal part of the simulation domain. **G–J** Variation of domain thickness (hp/h = 0.3 in **H** and **J**) also transforms simpler two-component continuous RD models into discrete systems (**H**, **J**). Profiles of average concentrations (at the y coordinate indicated by the white dotted lines) are shown; shades in (**H**, **J**) are minimum-maximum concentration intervals in the z direction.

scales with realistic geometries. To this end, we used optical high-resolution episcopic microscopy (HREM[38]) to reconstruct the exact 3D geometry of a patch of skin, of size 4 by 8 scales, sampled from a young adult ocellated lizard. The sample was fixed, dehydrated and then infiltrated and embedded in a rigid resin dyed with eosin. The block was then sectioned with an HREM device including a motorised microtome and a digital camera mounted on a microscope for automated image acquisition of successively-exposed block surfaces. We then reconstructed the 3D geometry of the RD domain by using the intrinsically-aligned images for volume rendering. As xanthophores and iridophores occupy the top portions of the dermis, while melanophores are able to reach somewhat deeper zones of the skin (especially in green scales[27]), one could argue that the simulation 3D field should be restricted to the region occupied by chromatophores (instead of using the whole skin thickness). Hence, we used the OpenCV computer vision library[39] to identify, in each of the HREM eosin-contrasted 2D images (Fig. 3A), the top ($S_t$) and bottom ($S_b$) surfaces of the skin, as well as pixels occupied by black pigments (Fig. 3B). We then plotted, for each

vertical line of pixels on all sections, the average position of black pixels against the thickness ($T_S$) of the skin (Fig. 3C). This procedure allows to easily distinguish green and black scales characterised, respectively, by low and high average positions of black pixels in the skin. We then estimated the portion of the skin populated by melanocytes by fitting a fourth-order polynomial to the lower bound of the three-standard deviation (3-std) interval around the average pigment position for all scales. Hence, this analytical expression defines the lower boundary of the domain occupied by chromatophores (Fig. 3D) in the ocellated lizard's actual skin. Continuous RD numerical simulations (with the same RD model parameters as in Fig. 2A–F) in that restricted 3D domain (Fig. 3E) or in the whole skin (Fig. 3F) both generate discrete steady-state patterns with a scale-by-scale colouration, confirming that (i) the actual geometry of the ocellated lizard skin is sufficient for the phenomenon to occur and (ii) this scale colour discretisation is robust to variations of the ratio between inter-scale and scale skin thicknesses. Note that HREM does not allow to reconstruct the geometry of a skin sample more than about 1 to 2 cm. To overcome the limitation of performing 3D

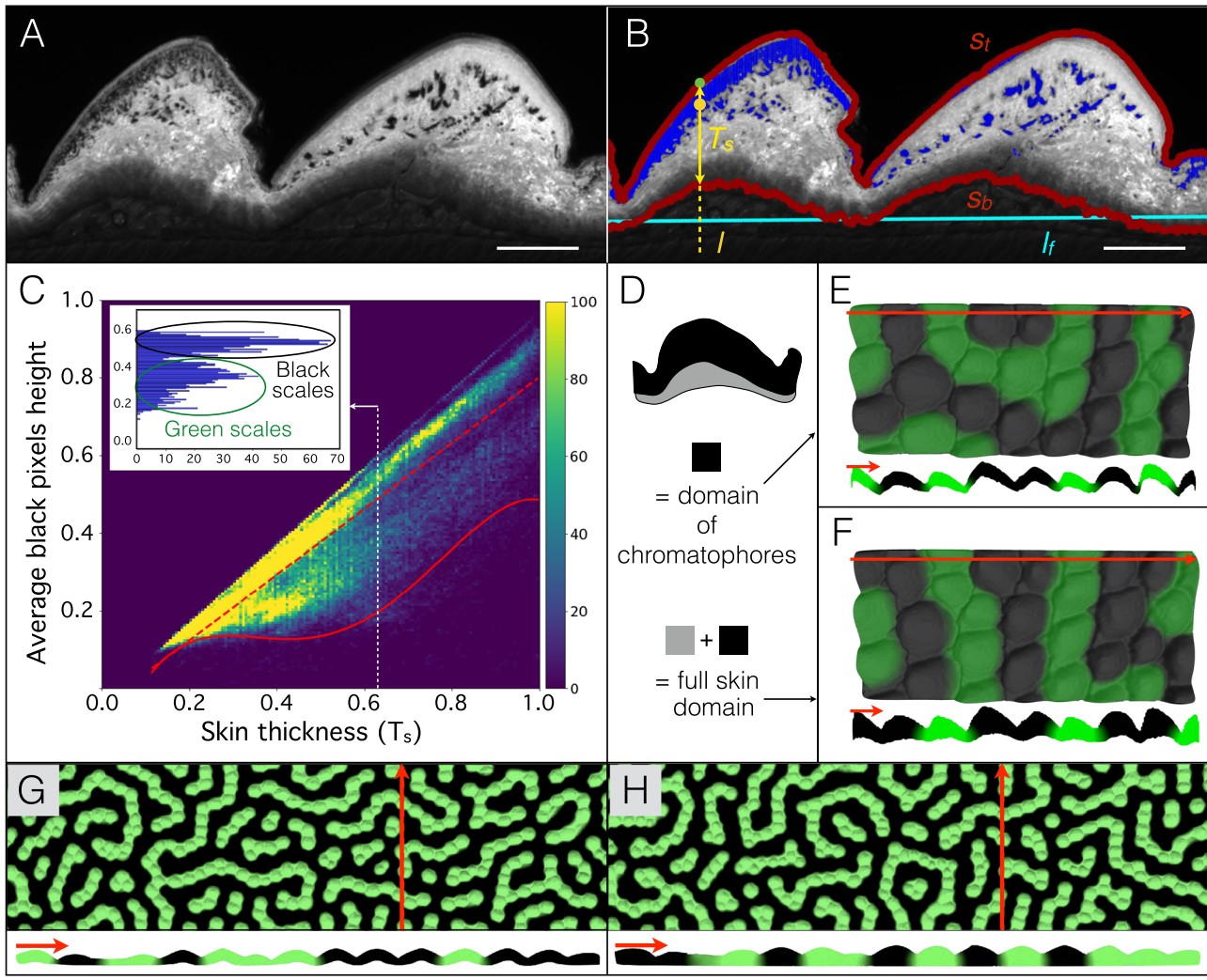

**Fig. 3 Actual skin geometry generates a discretised pattern in ocellated lizards. A** HREM section in a black scale (left) and a green scale (right). Scale bar = 200 μm. **B** Automated detection of top ($S_t$) and bottom ($S_b$) skin surfaces (red) and of black pixels (marked in blue) corresponding to melanin in melanophores. Cyan line = linear fit ($l_f$) to $S_b$. For each point on the top surface (e.g., green spot), skin thickness (yellow double-headed arrow, $T_s$) and average height (yellow spot) among all black pixels falling on the line $l$ perpendicular to $l_f$ were computed. **C** Black pixel height versus skin thickness in HREM sections from four black and four green skin scales (>250,000 data points, 150 bins in each direction). Red dashed line = linear fit to average pigment height. Plain red line = fourth-order polynomial fit to lower bound of 3-STD interval around the average pigment position. Inset: histogram for $T_s$ = 0.63 revealing two peaks corresponding to black and green scales. Height and thickness are normalised by the maximum observed value. **D** Diagram of a skin scale with full skin and chromatophore-restricted domains indicated. **E**, **F** 3D numerical simulations in the domain restricted to melanophore (**E**) or in the full skin (**F**) both generate a scale-by-scale steady-state colour pattern. Red lines indicate positions of the corresponding sections. **G**, **H** 3D numerical simulations in extended network of scales; surface geometry was inferred by shape-from-shading scanning of a large patch of lizard skin whereas chromotophore-restricted 3D domain (**G**) or full skin domain (**H**) was generated by applying statistics from HREM data.

simulations in such small lattices of 3D scales, we applied the skin thickness statistics obtained by HREM to a large patch of skin whose surface geometry was acquired using a shape-from-shading approach[40,41] (Supplementary Fig. 3 and see Methods). Numerical simulations using these reconstructed large lattices of 3D skin, both with a chromatophore-accessible domain (Fig. 3G) and with a full skin domain (Fig. 3H), generate scale colour switching dynamics and steady-state patterns very similar to those observed in real ocellated lizards.

**Dimensionality-reduction is accurate and computationally efficient.** Fourth, given that one primary interest of numerical simulations lies in the possibility to perform multiple comparative analyses, e.g., to study the effects of varying RD model parameters and geometry, we propose computationally-efficient 3D-to-2D dimensionality-reduction approaches

inspired from the Fick–Jacobs[42] and Bradley[43] 2D to 1D methods, i.e., with or without a position-dependent effective diffusion coefficient (See Methods). In short, we assume dominance of transport in the plane of the skin (a likely reasonable assumption given the much smaller thickness of the skin in comparison to its two other dimensions) while skin thickness variation is integrated as a correction of the mean concentrations of the RD components in the corresponding 2D simulation node. These approaches, aim at combining the best of two worlds: the high-precision of using the actual 3D skin geometry and the low computational cost of 2D numerical simulations. We therefore quantified both the loss of accuracy and the gain in computational efficiency of the dimensionality-reduction methods over bona fide 3D simulations. Using a network of Gaussian bumps (Fig. 4A), we show that 2D-reduced simulations produce steady-state

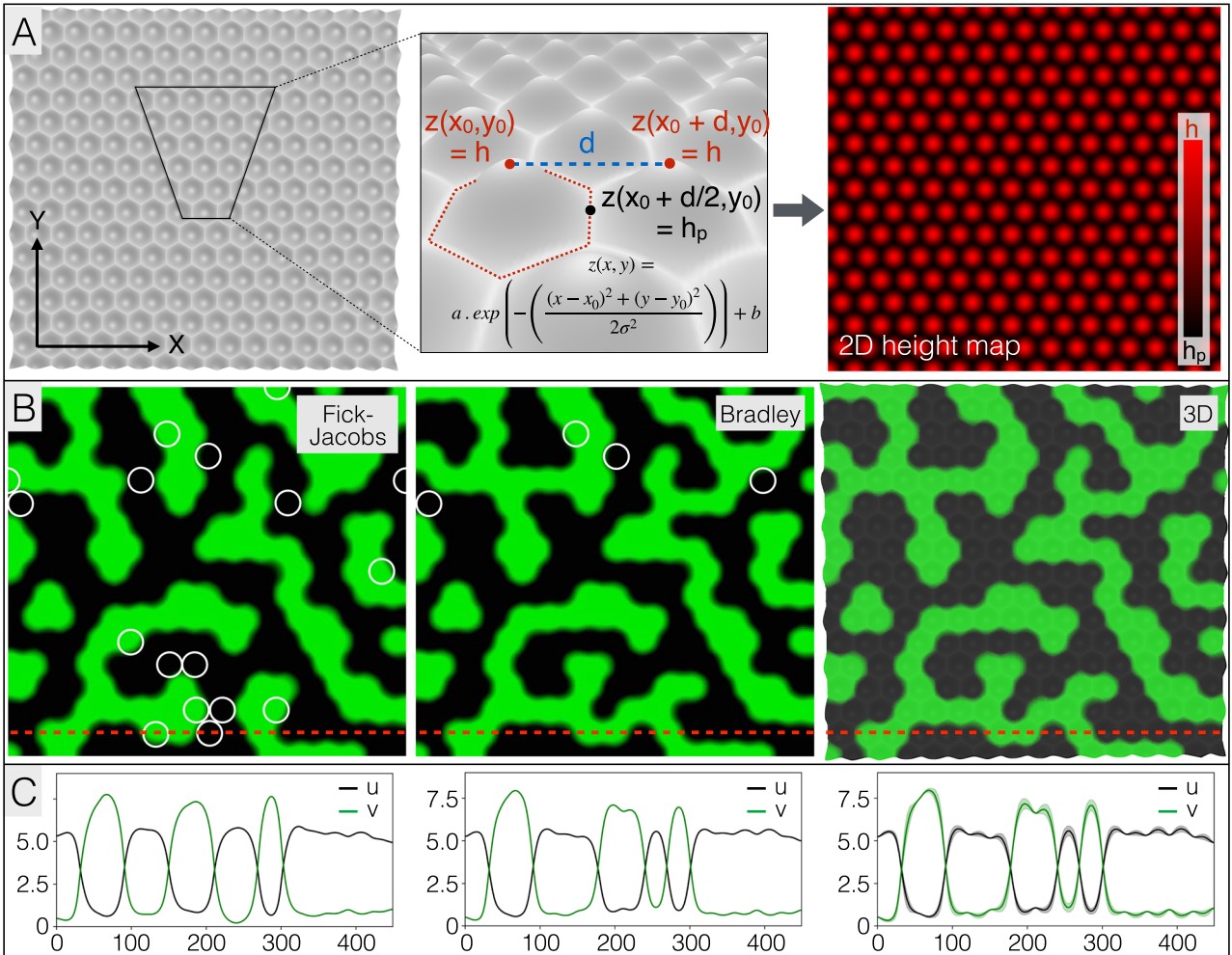

**Fig. 4 Dimensionality-reduction is accurate and computationally efficient. A** A parametrised hexagonal lattice of 3D scales (two left panels) is generated by using, for each scale, the two-dimensional Gaussian function (inset) with heights of scales centres $h$ and scales borders heights $h_p$; the domain surface geometry is produced by fitting the constants $a$ and $b$ of the Gaussian function and keeping the standard deviations identical in the X and Y directions ($\sigma_X = \sigma_Y$, symmetrical Gaussian bumps). The height information (right panel) is then used to perform 2D simulations with two different dimensionality-reduction approaches. **B** Starting with identical initial conditions, 2D-reduced simulations (with $h_p/h = 0.31$) on a lattice of $450 \times 417$ elements with the Fick–Jacobs (left panel) and Bradley (central panel) methods, as well as 3D simulations (right panel), produce qualitatively very similar steady-state scale-by-scale patterns. Individual scales (white circles) with colour states opposite to those observed in the bona fide 3D simulation (green or black instead of black or green) amount to 6.4 and 1.5% of the scales for the Fick–Jacobs and Bradley methods, respectively. Parameter values are $\varepsilon = 3$, $\sigma = 8$, $h = 16$, $D_u = D_v = 11.25$ and $D_w = 135$. **C** Profiles of components' concentrations are shown for the y coordinate indicated by the red dotted lines in the corresponding panels in (**B**); shades in right panel are minimum-maximum concentration intervals in the z direction (plain line is the average).

patterns that are qualitatively very similar to that of a corresponding 3D simulation with the same initial condition (Fig. 4B). In terms of accuracy, the 2D simulations based on the Fick–Jacobs and Bradley methods of dimensionality reduction produced patterns displaying respectively 6.4 and 1.5% of the scales with the wrong colour (black instead of green or vice versa). This result is confirmed when comparing the two methods with 3D simulations for a wide range of $h_p/h$ values (from 0.06 to 0.56): for each simulation, we project on a 2D plane the average steady-state green colour intensity (see Methods) across the z direction of each $(x_i, y_i)$ position in the 3D domain and subtract the image produced at steady state by one or the other dimensionality-reduction method. These comparisons indicate that 2D simulations are 10 to 20 times faster than 3D simulations (typically, 2 min instead of 30 min), but produce an average 7.0 and 2.2% of colour difference per simulation element, for the Fick–Jacobs and Bradley's methods, respectively.

**Growth affects patterning dynamics.** Fifth, we quantitatively compare the theoretical and observed effect of growth on the dynamics of the skin pattern length-scale variation. Indeed, in principle, the length scale of a RD system is invariant because it is determined by the diffusion coefficients and reaction rates. Although one could imagine that these coefficients and rates vary with the age of animals, the most parsimonious conjecture is that, as an animal grows, new geometrical elements of similar type (e.g., spots in a spotted pattern, stripes in a striped pattern) are introduced to maintain a pattern length scale of constant absolute value, i.e., a pattern of decreasing length scale relative to the size of the animal. This prediction of relative period doubling was observed in the *Pomocanthus semicirculatus* and *P. imperator* angel fishes[44]: in juveniles of both species, new stripes appear between pre-existing semi-circular stripes as the animal is growing. Similarly, in the second species, after the juvenile pattern transforms into a sub-adult set of horizontal stripes, additional stripes are added, as the animal continues to

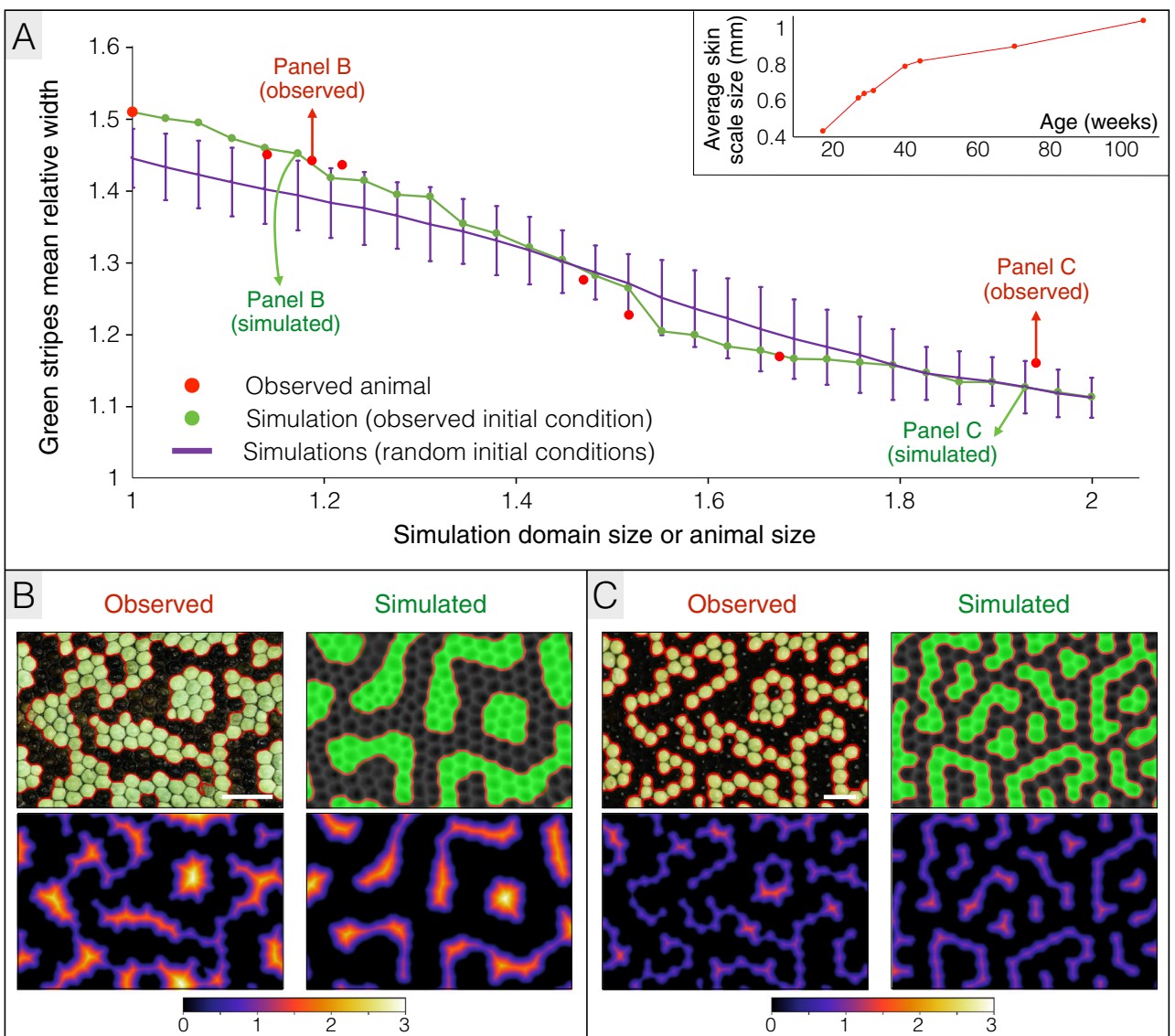

**Fig. 5 Growth affects patterning dynamics. A** Growth (inset: growth curve of the animal) causes a monotonous decrease in the mean relative length scale of the pattern both in the real animal (red spots = 18, 28, 30, 32, 41, 45, 71 and 106 week-old animal) and in a quasi-static growth simulation (green line; initial condition = 18-week-old observed pattern). The purple line shows the time evolution of the pattern mean length scale averaged among 20 independent simulations starting form different random initial conditions (30% randomly-distributed black skin scales); error bars = 3 standard deviation intervals. Simulations were performed in 2D by applying dimensionality reduction on the 3D chromatophore domain (Fig. 3D). The size of simulation domains is normalised to the skin patch size at 18 weeks. **B**, **C** Comparisons between real and simulated patterns for normalised domain sizes of ≈1.15 (**B**) and ≈1.95 (**C**). Top panels show the patterns with detected borders (indicated in red) and bottom panels show the distance (expressed in average skin scales) of each green scale to the border. The observed and simulated patterns both exhibit green labyrinthine stripes that are 2–3 and 1–2 skin scales wide in **B** and **C**, respectively. Scale bars = 2 mm. Time evolution of the pattern length scale is also illustrated in Supplementary Movie 1.

grow, through a series of stripe extensions and branching-point displacements[44].

Here, we use numerical simulations to analyse, in ocellated lizards, how growth is expected to affect the dynamics of labyrinthine pattern development, especially the length scale of the pattern. Using the most-accurate dimensionality-reduction approach described above, we use a large patch of dorsal skin to perform RD simulations in a growing 3D domain. Making the reasonable assumption that RD equilibrium is reached much faster than the skin is growing, we use a quasi-static technique where the simulation domain is grown, and corresponding dimensionality-reduction parameters are updated, incrementally each time RD steady state is attained (see Methods). Using the pattern of a 18 weeks old ocellated lizard (i.e., around the

transition between the juvenile brown pattern and the subadult green and black pattern) as initial condition, and the growth curve derived from that same animal (inset of Fig. 5A), the simulation generates an average relative width of green labyrinthine stripes (i.e., the mean relative length scale of the pattern measured in number of scale diameters; Supplementary Fig. 4) that monotonously decreases by 27% as the simulation domain grows (green line in Fig. 5A and Supplementary Movie 1). Moreover, the average curvature of the pattern borders increases by 40% (green line in Supplementary Fig. 5), indicating that the pattern is becoming more labyrinthine as the simulation domain grows. Remarkably, these dynamics are quantitatively highly similar to those observed in the corresponding real ocellated lizard (red dots in Fig. 5 and Supplementary Fig. 5), suggesting

that the observed length-scale reduction and 'labyrinthinisation' of the pattern are exclusively due to growth and not to age-related variation of the dynamical system parameter values. Note that the phenomenon linking growth, pattern length scale and curvature is robust to variation of initial conditions as the relative length scale of the pattern consistently decreases by about 25%, and the average border curvature consistently increases by 34%, in 20 independent simulations (purple line in Fig. 5 and Supplementary Fig. 5) starting with different initial patterns of 30% randomly-distributed black scales (i.e., the observed approximate proportion of black scales at 18 weeks of age).

These results also indicate that, despite growth, the ocellated lizard body scales remain smaller (even in the adult) than the length scale of the RD pattern in the plane, hence, each body scale takes a uniform colour. To obtain a pattern within scales, the diffusion coefficients should be smaller or scales should be larger. The second possibility does exist in ocellated lizards: larger tail scales are often dichromatic (Supplementary Fig. 6A). Our RD model is compatible with this observation as it produces a pattern within tail scales whose genuine 3D geometry was reconstructed with HREM (Supplementary Fig. 6B, C).

## Discussion

Assuming that a substantial reduction of RD diffusion coefficients at the borders of ocellated lizard's polygonal 2D scales is justified by spatial variation of the actual skin geometry in the third spatial dimension (i.e., the presence of thin skin in between thicker scales), we previously demonstrated[27] that this ad hoc parameter transforms a Turing continuous RD system into discrete macroscopic von Neumann CA dynamics that computes a labyrinthine skin pattern of black and green scales. These dynamics are highly similar to the actual post-hatching self-organisational skin patterning process observed in that species[27]. Here, we quantitatively validate the above-mentioned conjecture by confirming that variation of skin thickness in bona fide RD numerical simulations (with constant diffusion coefficients) performed in realistic 3D domains produces a scale-by-scale colouring and a CA behaviour. We additionally show that this phenomenon is robust to variation of the RD model and that growth of the animal considerably and realistically affects the patterning dynamics. Given that the current histological and episcopic microscopy data did not evidence any scale substructure (beyond the thinning of the skin per se) that could locally affect the diffusion constants in the inter-scale skin, the data and analyses presented here demonstrate that the mere superposition of the lizard's 3D skin geometry with a continuous RD system generates a CA.

The diffusion terms in the RD PDE can represent a series of biological processes: effective diffusion of signalling molecules, cell movements and/or the production of long-range cellular projections. If one makes the reasonable assumption that the fundamental molecular and cellular mechanisms governing the self-organising skin colour patterning process are similar in ocellated lizards and zebrafish, the extensive literature on the latter (e.g.[45–47]) indicates that some long-range interactions among chromatophores are mediated by long cell projections (called 'airinemes'[48]). Note that, more generally, such signalling filopodia can mediate morphogen transport and gradient establishment in rapidly-expanding tissues[49], such as in early developmental processes. We think that the geometric constrictions constituted by the inter-scale skin is very likely to affect not only diffusing molecules and chromatophore movements but also airineme transport. Indeed, recent findings indicate[11] that airinemes production and delivery require a macrophage relay: after recognising surface blebs of xanthoblasts, macrophages drag

airineme vesicles and deliver them to melanophores, providing the long distance communication between these two chromatophore types. These data help understand why the RD framework is efficient in recapitulating the effective CA process in the ocellated lizard as the wandering of the macrophages with the xanthophores' cellular long projections is (i) likely affected by the small thickness of the inter-scale skin restrictions and (ii) might explain a diffusion-like dissemination of long-distance signals during the skin colour patterning process.

Ever since Alan Turing[15], RD equations have been used as a paradigmatic model of pattern development[19,21,27,50]. The success of the Turing model is for multiple reasons: (i) the combination of nonlinear reactions with short- and long-range diffusion makes this framework very simple and mathematically elegant, (ii) the system of PDEs is often locally tractable through linearisation, (iii) it can generate a vast array of behaviours, including a diversity of stationary patterns and (iv) the cell–cell interactions in real biological systems occur through effective diffusion[51] of morphogens in tissues and/or through short- and long-range contacts that can be translated into small and large diffusion coefficients. Although the combination of multiple interactions into a few parameters can be viewed as a strength of RD approaches, other scholars would argue that it makes clear biological interpretation of these parameters difficult. For example, one of the three components in the Nakamasu et al. model[21] very efficiently accounts for long-range interactions, but it is unclear what its biochemical correspondence is in the actual biological system.

One potential solution to this issue is the use of AB models[24] that simulate the movements and interactions of cells as autonomous agents. Although it remains to be determined if both RD and AB systems can always model the same set of interactions, one could argue that AB models can more readily incorporate newly-identified interactions and/or cell types, especially in species amenable to cell biology investigations (e.g., the zebrafish). For instance, iridophores, recently recognised as playing a role in the zebrafish skin colour patterning[45–47], and absent from the RD model of Nakamasu et al.[21], have been successfully incorporated in AB models[22,23]. Note however, that AB models are not immune to simplification: parameters are typically fitted to the expected pattern and dynamics[24] and models incorporate some level of idealisation of the particles that indirectly take into account cell behaviours[22]. Furthermore, the computational cost scales with the simulation domain size less favourably for AB models than for RD models. Indeed, agents are the intrinsic discretisation level of AB models independently of the simulation domain size, whereas the discretisation size of domain nodes can be adapted in RD models relative to the pattern length scale and the full domain size. Finally, it is worth noting that diffusion terms of some RD models can be substituted by a small number of biologically-explicit parameters (such as cell–cell adhesion) and produce continuous models[52,53] that can more easily be interpreted, and modified, in terms of actual biochemical quantitative measurements.

The availability of a diverse set of modelling approaches is a benefit to the community as it allows for the cross-validation of the significance of specific parameters in specific observations. In this respect, the observation of a scale-by-scale colour patterning in our 3D RD simulations, using three-components but also two-component models, is a strong indication that our conjecture (3D skin scales generating a discretisation of the skin colour pattern) is general, i.e., it might not depend on the nature (RD or AB) of the model implementing the microscopic interactions. In other words, we are rather confident that a 3D AB model (to the best of our knowledge, all current implementations modelling skin colour patterning are in 2D) would recapitulate our results.

On the other hand, the precise modelling of other aspects of skin colour patterning in the ocellated lizard is likely to require substantial cell biology investigations in that species, a possibly unreachable endeavour with current technologies. In a sense, the use of simpler RD models, instead of more elaborated AB models, is justified by the fact that all cellular mechanisms involved in the patterning process are not yet understood. Note also that, although the fit between observations and simulations (both with CA and RD models) is spectacular in statistical terms[27], the exact prediction of the adult pattern could be improved by using more subtle initial conditions, i.e., real colours instead of assigning all scales to one of two sets (corresponding to 'green' and 'black') of specific $(u,v,w)$ values followed by the simple addition of ±1% random noise (see Methods).

Given that our central argument in the present work is that 3D geometry of the domain, in which cell–cell interactions occur, is sufficient to explain the transformation of a microscopic system of reaction and diffusion into a discrete CA at a much larger scale (the spatial scale of skin scales), one could argue that one questionable approximation in our 3D simulations is that we estimate the thickness of the domain in terms of either (i) the (spatially-variable) absolute total thickness of the skin or (ii) a 'chromatophore domain' estimated from the distribution of melanin in all scales (Fig. 3D). The first setting is indeed an approximation because it is unlikely that chromatophores interact with each others across the whole depth of the dermis. The second setting is also an approximation because it homogenises the differences that exist between black and green scales (Fig. 3C). However, fitting the lower bound of the domain separately for green and black scales would not help because scales change colour, i.e., we would need instead to dynamically change the thickness of the domain in each scale that shifts from green to black or black to green. A proper assessment of such a domain size variation dynamics would require sampling scales at different stages of the colour switching process to quantitatively assess the correspondence of scale colour with chromatophore distribution across the dermis depth. Note that, difficulty of experimental realisation aside, such an approach would anyway remain an approximation as the distribution of pigments observed with histology does not equate to the distribution of cells. Indeed, it is likely that the large amount of pigments at the top of black scales is due to melanophores sending projections towards the dermis/epidermis interface rather the cells moving themselves. In other words, we do not have reliable data justifying the introduction of an additional parameter that would define the local thickness of the domain as a function of u and/or v. However, most importantly, our simulations generate natural scale colour switching dynamics and realistic steady-state patterns regardless whether the 3D domain corresponds to the full skin thickness (Fig. 3F, H) or is restricted to the estimated part of the skin accessible to chromatophores (Fig. 3E, G), suggesting that the phenomenon of scale colour discretisation is robust to variations of the ratio between inter-scale and scale skin thicknesses.

One difficulty in comparing past and current modelling of the ocellated lizard skin colour patterning process is to relate the parameters that generate a scale-by-scale coloration and a CA behaviour in 2D[27] versus 3D simulations, i.e., the reduced diffusion coefficient ($P < 0.2$) in the former and the relative thinning of the inter-scale skin ($h_p/h < 0.3$) in the latter. A quantitative comparison is challenging for multiple reasons. First, the width of border scales is minimal (i.e., one discretisation element) in 2D simulations[27], discrete in 3D prism lattices (T in Fig. 2D) but continuous for 3D Gaussian bumps and realistic geometries derived from episcopic microscopy. However, sensitivity analyses indicate that the progressive thinning of inter-scale skin generates the same qualitative transitions of the pattern, in both 2D (ref. [27])

and 3D simulations (Supplementary Fig. 1): from unaffected by scale boundaries (right inset in Supplementary Fig. 1) to scale-by-scale, with motifs that are first 2 to 3 scales wide (central inset) then 1 or 2 scales wide (left inset).

**On growth and form.** Our previous study[27] in the ocellated lizard (i) identified that the dynamics of skin colour development are produced by a probabilistic CA (made of skin scales) that computes the adult labyrinthine steady-state pattern, indicating that this family of discrete computational systems[28–31], long-thought to be purely abstract models, have been generated by biological evolution; and (ii) suggested that the scale-by-scale coloration and the CA dynamics of pattern development both emerge from the underlying microscopic system of cell interactions (modelled by a continuous RD system) superposed to the geometry of the skin—i.e., thinning of the skin in between scales causes the discretisation of the continuous (microscopic) RD dynamics at a larger (meso-/macroscopic) spatial scale: the (spatial) scale of (skin) scales.

However, that study was entirely based on the untested assumption that the spatial variation of thickness of the lizard's skin translates into a substantial reduction of RD diffusion coefficients at the position of scale borders[27]. Here, we have tested this assumption by using a bona fide 3D RD model, i.e., RD simulations are performed in a 3D domain that mimics the geometry of the lizard skin. Using, first idealised lattices of 3D hexagonal prisms and, second, realistic quasi-hexagonal lattices of 3D scales generated from biological (histological) data on ocellated lizard skin, we now show that geometry itself, rather than an artificial reduction of diffusion coefficients at scale borders, is sufficient for causing the emergence of a scale-by-scale coloration and cellular-automaton dynamics. In addition, using RD models vastly different from that of Nakamasu et al.[21], we show that the phenomenon (geometry causing the emergence of a macroscopic discretised dynamics) is likely to be very general, i.e., RD-model independent.

Because of the high computational intensity of 3D numerical simulations (despite their porting on GPUs), we then investigated dimensionality-reduction approaches that project in 2D the actual 3D geometry in the form of RD components concentrations and effective diffusion coefficients that are both position-dependent on the 2D simulation domain. Our results indicate that these methods are 10 to 20 times faster than 3D simulations and generate relatively low errors.

Finally, as a tribute to D'Arcy Thompson[32], we show that animal growth substantially impacts form of the steady-state colour pattern by producing a monotonous and substantial decrease of the relative length scale, as well as an increase in the convolution, of the ocellated lizard skin colour labyrinthine pattern.

## Methods

**Animals and ethics statement.** Ocellated lizards were bred in Milinkovitch's laboratory, Department of Genetics and Evolution, University of Geneva, Switzerland. Maintenance of, and experiments on animals were approved by the Geneva Canton ethical regulation authority (authorisations GE/82/14, GE/73/16 and GE/27/19) and performed according to Swiss law. These guidelines meet international standards.

**3D reaction-diffusion numerical model.** We use the following system of non-linear PDEs developed for zebrafish[21] and ocellated lizards[27]:

$$\frac{\partial u}{\partial t} = F(u, v, w) - c_u u + D_u \nabla^2 u$$

$$\frac{\partial v}{\partial t} = G(u, v, w) - c_v v + D_v \nabla^2 v$$

$$\frac{\partial w}{\partial t} = H(u, v, w) - c_w w + D_w \nabla^2 w$$

where

$$F(u, v, w) = \begin{cases} 0 & : c_1 v + c_2 w + c_3 < 0 \\ c_1 v + c_2 w + c_3 & : 0 \leq c_1 v + c_2 w + c_3 \leq F_{\max} \\ F_{\max} & : F_{\max} < c_1 v + c_2 w + c_3 \end{cases}$$

$$G(u, v, w) = \begin{cases} 0 & : c_4 u + c_5 w + c_6 < 0 \\ c_4 u + c_5 w + c_6 & : 0 \leq c_4 u + c_5 w + c_6 \leq G_{\max} \\ G_{\max} & : G_{\max} < c_4 u + c_5 w + c_6 \end{cases}$$

$$H(u, v, w) = \begin{cases} 0 & : c_7 u + c_8 v + c_9 < 0 \\ c_7 u + c_8 v + c_9 & : 0 \leq c_7 u + c_8 v + c_9 \leq H_{\max} \\ H_{\max} & : H_{\max} < c_7 u + c_8 v + c_9 \end{cases}$$

The variables $u$ and $v$ are the densities of the short-range factors (corresponding to melanophores and xantophores, respectively) and the variable $w$ is the long-range factor activated by $u$ and inhibited by $v$. The signs of the variables $c_1$, $c_2$, $c_4$, $c_5$, $c_7$ and $c_8$ are constrained by the cell interaction model discussed elsewhere[21,27]. The variables $c_u$, $c_v$ and $c_w$ are decay terms. Diffusion coefficients ($D_u = D_v \ll D_w$) represent the short and long range of cell interactions. The parameters used in our simulations are $c_1 = -0.04$, $c_2 = -0.056$, $c_3 = 0.382$, $c_4 = -0.05$, $c_5 = 0$, $c_6 = 0.25$, $c_7 = 0.016$, $c_8 = -0.03$, $c_9 = 0.24$, $c_u = 0.02$, $c_v = 0.025$, $c_w = 0.06$, $D_u = D_v = 1.125$, $D_w = 12 \cdot D_u = 13.5$, $F_{\max} = G_{\max} = H_{\max} = 0.5$, $dt = 0.01$, unless stated otherwise. Parameter values were tuned to obtain ocellated lizard patterns. All signs of the reaction-term coefficients are identical, and their absolute values are similar, to those in Nakamasu et al.[21], suggesting that cell–cell interactions are comparable in the ocellated lizard and the zebrafish. The intensity of green $g \in [0,1]$ for every simulation node was calculated using:

$$g = \min(\max((v - u)/(v + u) + 0.5, 0), 1).$$

This relation insures symmetry between the widths of green and black stripes within the single wavelength of the pattern. Using a non-symmetrical scheme (e.g., by using a value other than 0.5 in the max relation) would change the relative widths of green and black stripes but would not change the wavelength of the full pattern (i.e., the succession of a green + a black stripe). Simulations are performed on a regular network of spacing $\varepsilon$ with periodic or no-flux boundary conditions at the domain borders. Concentrations at all nodes are updated using the Euler forward method and the Laplacian term is approximated using a seven-point stencil discrete operator:

$$\nabla^2 u \approx \frac{u_{\text{north}} + u_{\text{south}} + u_{\text{west}} + u_{\text{east}} + u_{\text{top}} + u_{\text{bottom}} - 6u}{\varepsilon^2}$$

Simulations are stopped when each node satisfies:

$$|u(t - \Delta t) - u(t)| + |v(t - \Delta t) - v(t)| + |w(t - \Delta t) - w(t)| < 10^{-7}$$

Implementation was optimised by exploiting the parallelisation capacity of both GPU and CPU based on a previously published numerical scheme[34]. At the GPU level, concentrations are stored in global memory and every computation of a time step is assigned to a kernel function (i.e., threads compute individual grid points). To avoid repeated reading from the slow global memory, the data is copied at every time step to a shared memory of a thread block. As calculation of the stencil requires an extra layer of grid points around it (to allow computation of all points in the block), blocks that overlap in every direction are created, i.e., all threads read their corresponding grid points and fill the shared memory whereas grid points on the extra layer do not compute. Parallelisation at the CPU level consists in two synchronised threads: one thread is dedicated to both GPU kernel invocation and transfer of the data from the GPU to the CPU memory, while the other writes the data.

Initial conditions for simulations with hexagonal prisms (Fig. 2A–F) are random assignments of each prism to green $(u, v, w) = (1.2, 6.6, 2.3)$ or black $(u, v, w) = (5.3, 0.92, 4)$ followed by addition of ±1% random noise to each simulation node. Boundary conditions are periodic in x and y, and no-flux at the domain top and bottom borders. Simulations in the HREM-reconstructed 3D domain were performed with 250 elements in the horizontal direction, $\varepsilon = 0.9$, and uniform initial condition with ±1% random noise.

**Two-component RD models.** We tested the potential general effect of the skin 3D geometry on the discretisation of RD systems at the spatial scale of skin scales by using the Grey-Scott and the Schnakenberg two-component RD models, i.e., systems of nonlinear PDEs very different from, and simpler than, that used to model the zebrafish and ocellated lizard pattern development dynamics[21,27]. The Grey-Scott model[35] corresponds to the following system in dimensionless units[36]:

$$\frac{\partial u}{\partial t} = -uv^2 + F(1 - u) + D_u \nabla^2 u$$
$$\frac{\partial v}{\partial t} = uv^2 - (F + k)v + D_v \nabla^2 v \tag{1}$$

whereas the Schnakenberg model[37] corresponds to:

$$\frac{\partial u}{\partial t} = \gamma(a - u + u^2 v) + \nabla^2 u$$
$$\frac{\partial v}{\partial t} = \gamma(b - u^2 v) + d \nabla^2 v \tag{2}$$

Boundary conditions in Fig. 2G–J are periodic in x and y, and no-flux at the domain borders in the z direction. RD parameters for Grey-Scott are $\varepsilon = 0.1$, $F = 0.042$, $k = 0.063$, $D_u = 0.15$ and $D_v = 0.075$. RD parameters for Schnakenberg are $a = 0.1$, $b = 0.9$, $\gamma = 1$, $d = 40$. Initial conditions in Fig. 2G and I are random assignments of each hexagonal prism to green $(u, v) = (0.5, 0.25)$ or black $(u, v) = (0, 1)$ followed by addition of ±1% random noise to each simulation node. Initial conditions in Fig. 2H and J are the steady-states of Fig. 2G and I.

**Episcopic microscopy and extending statistics to large simulation domains.** A dorsal patch of skin of about $1 \times 0.5$ cm (8 by 4 scales) was sampled from a euthanised young adult ocellated lizard and fixed overnight in 4% paraformaldehyde (PFA) at +4 °C. The sample was then washed in phosphate-buffered saline (PBS) for 1 h, dehydrated through a series of methanol solutions in water (1 h in each solution of 30, 50, 70, and 100% methanol) and stored at −20 °C in fresh 100% methanol. For HREM, the skin sample underwent fluorescent staining and embedding in Technovit 8100 resin (Kulzer, Germany): it was washed twice in acetone at +4 °C for 1 h and incubated in Technovit infiltration solution with 2 mg/ml of Eosin B and 0.5 mg/ml Acridine Orange (Sigma-Aldrich, USA) for 5 h at +4 °C. A second infiltration with fresh solution was then performed in the same conditions. After withdrawal from the infiltration solution, and removal of excess solution with filter paper, the sample was oriented (to insure sectioning perpendicular to the skin surface) in custom cuvettes and mounted in pre-cooled Technovit polymerisation solution containing 6 mg/ml Orasol Black (Sigma-Aldrich, USA) for overnight polymerisation at +4 °C. The resulting block was sectioned at 3.5 μm thickness with an OHREM sectioning/imaging system (Indigo Scientific, UK). A stack of 1136 high-resolution images ($4080 \times 3072$ pixels, 16 bits) were obtained using the red fluorescent protein filters set. These images were then used to produce the 3D simulation domain (defined by either the whole skin or the skin portion occupied by melanophores; Fig. 3D) in which numerical simulations were performed (Fig. 3E, F).

The statistics of thickness variation derived from the HREM-reconstructed skin were then applied to a large patch of skin as follows. The high-frequency surface micro-geometry of a dorsal skin patch of about 700 scales was acquired on a 18-week-old animal using a shape-from-shading approach[40] implemented in a robotic high-resolution system (R²OBBIE-3D; ref. [41]). After the reconstructed patch was aligned to the xy plane, its height was thresholded, generating a series of disconnected convex sets whose centroids represent the positions of scales. The average distance between the centroids of neighbouring skin scales $d_{\text{avg}}^M$ was computed and the surface was isotropically scaled by a factor $d_{\text{avg}}^H / d_{\text{avg}}^M$, where $d_{\text{avg}}^H$ is the average distance between neighbouring skin scale centres on the surface of the HREM-reconstructed sample. Then, the bottom surface of the simulation skin domain was generated using the following expression:

$$z_{\text{bottom}}^{\text{skin}}(x, y) = z_{\text{top}}^{\text{skin}}(x, y) - \text{norm}_{\text{HREM}}(z_{\text{top}}^{\text{skin}}(x, y)) \cdot \max_h^{\text{HREM}} \tag{3}$$

where $z_{\text{top}}^{\text{skin}}$ is the top surface of the skin and

$$\text{norm}_{\text{HREM}}(z) = \frac{z - \min(z_{\text{top}}^{\text{skin}})}{\max(z_{\text{top}}^{\text{skin}}) - \min(z_{\text{top}}^{\text{skin}})} \left(1 - \frac{\min_h^{\text{HREM}}}{\max_h^{\text{HREM}}}\right) + \frac{\min_h^{\text{HREM}}}{\max_h^{\text{HREM}}} \tag{4}$$

is a function mapping all z values to the interval $[\min_h^{\text{HREM}} / \max_h^{\text{HREM}}, 1]$. Note that $\min_h^{\text{HREM}}$ and $\max_h^{\text{HREM}}$ represent the minimum and the maximum unnormalised skin height in the HREM patch. As chromatophores do not reach the deepest zones of the skin, the domain occupied by chromatophores (Fig. 3D), i.e., the volume probably more valid than the whole skin domain for 3D numerical simulations, was reconstructed using the following expression:

$$z_{\text{bottom}}^{\text{chromatophores}}(x, y) = z_{\text{top}}^{\text{skin}}(x, y) - \left(\text{norm}_{\text{HREM}}\left(z_{\text{top}}^{\text{skin}}(x, y)\right) - p_{-3\text{std}}\left(z_{\text{top}}^{\text{skin}}(x, y)\right)\right) \cdot \max_h^{\text{HREM}} \tag{5}$$

where $p_{-3std}$ is the fourth-order polynomial fit to the lower bound of the 3-std interval around the average pigment position for all scales.

**Dimensionality-reduction.** Dimensionality reduction from 2D to 1D for diffusion in a narrow 2D channel has been formally derived[42,43]. To our knowledge, derivation of dimensionality reduction from 3D to 2D does not exist in the literature. Here, we propose computationally-efficient dimensionality-reduction approaches where skin thickness variation is projected in 2D as tuned RD components' concentrations and effective diffusion coefficients. More specifically, for a RD system of the form:

$$\frac{\partial c}{\partial t} = f(c) + D \vec{\nabla}^2 c \tag{6}$$

in a 3D geometry bounded (i.e., with no-flux) at bottom and top by, respectively, two functions $z = z_1(x, y)$ and $z = z_2(x, y)$ (Supplementary Fig. 1A), the integrated

concentration $c(x, y, z, t)$ with respect to the z coordinate is given by:

$$Q = Q(x, y, t) = \int_{z_1(x,y)}^{z_2(x,y)} c(x, y, z, t)dz \qquad (7)$$

Given that the variation in z of u, v, and w at any (x,y) point is much smaller than the difference of these concentrations between green and black scales (as shown in bona fide 3D simulations; Fig. 2F), we assume that there is no variation of the 3D density $c$ in the z direction. This assumption is justified, for diffusion without reaction terms, using the systematic expansion of the diffusion equation at leading order[43] (and its straightforward generalisation to higher dimensions). When adding the reaction terms, the assumption of no variation of $c$ in the z direction remains justified by the thickness of the domain being smaller than the typical length-scale of the RD pattern. This assumption then results in:

$$Q \approx c(x, y, t) \cdot z(x, y) \qquad (8)$$

where $z(x, y) = z_2(x, y) - z_1(x, y)$ is the domain height.

For an elementary column volume $V \approx \epsilon^2 z(x, y)$ (Supplementary Fig. 1B) in a RD system, the general mass conservation law requires:

$$\frac{\partial}{\partial t} \int_V c dV = -\oint_S \vec{J} \cdot \vec{n} dS + \int_V f(c) dV \qquad (9)$$

where the first and second right-hand side terms are the diffusion and reaction terms, respectively. $\vec{J} = -D\vec{\nabla}c = (J_x, J_y, 0)$ is the diffusional flux per unit area accounting for a constant concentration in the z direction, $S$ is the surface area of the element and $\vec{n}$ is the outward facing surface normals. Given the assumption that $c$ does not vary in the z direction, we can write the left-hand side of the equation as:

$$\frac{\partial}{\partial t} \int_V c dV \approx \frac{\partial Q}{\partial t} \epsilon^2 \qquad (10)$$

Considering each side of the elementary volume separately, the diffusion term gives:

$$\oint_S \vec{J} \cdot \vec{n} dS = \int_{S_{x+}} J_x(x + \epsilon/2, y) dS - \int_{S_{x-}} J_x(x - \epsilon/2, y) dS$$
$$+ \int_{S_{y+}} J_y(x, y + \epsilon/2) dS - \int_{S_{y-}} J_y(x, y - \epsilon/2) dS$$
$$\approx \epsilon \cdot (J_x(x + \epsilon/2, y) \cdot z(x + \epsilon/2, y) - (J_x(x - \epsilon/2, y) \cdot z(x - \epsilon/2, y))$$
$$+ \epsilon \cdot \left( J_y(x, y + \epsilon/2) \cdot z(x, y + \epsilon/2) - (J_y(x, y - \epsilon/2) \cdot z(x, y - \epsilon/2)) \right) \qquad (11)$$

Whereas the reaction term becomes:

$$\int_V f(c) dV \approx f\left(\frac{Q}{z}\right) \epsilon^2 z \qquad (12)$$

Substituting these three latter expressions in the general mass conservation equation and dividing by $\varepsilon^2$, we obtain:

$$\frac{\partial Q}{\partial t} \approx \frac{(J_x(x - \epsilon/2, y) \cdot z(x - \epsilon/2, y) - (J_x(x + \epsilon/2, y) \cdot z(x + \epsilon/2, y))}{\epsilon}$$
$$+ \frac{\left( J_y(x, y - \epsilon/2) \cdot z(x, y - \epsilon/2) - (J_y(x, y + \epsilon/2) \cdot z(x, y + \epsilon/2)) \right)}{\epsilon} + f\left(\frac{Q}{z}\right) z \quad (13)$$
$$= \left( \frac{\partial}{\partial x}(J_x z) + \frac{\partial}{\partial y}(J_y z) \right) + f\left(\frac{Q}{z}\right) z$$

i.e.,

$$\frac{\partial Q}{\partial t} \approx \vec{\nabla} \cdot \left( Dz\vec{\nabla}\left(\frac{Q}{z}\right) \right) + f\left(\frac{Q}{z}\right) z \qquad (14)$$

In one dimension, the first right-hand side term is equivalent to the Fick–Jacobs equation[42] where $D$ is a constant diffusion coefficient. Thickness of the domain is therefore taken into consideration by the linear correction of the diffusion coefficient $D(x,y)$ by the corresponding value $z(x,y)$ at that position.

However, it has been suggested[43] that this model can be improved by replacing $D$ in the above equation by a position-dependant effective diffusion coefficient $D(x, y)$. For a narrow and asymmetric 2D channel of height $z(x)$ confined between two smooth functions $z_1(x)$ and $z_2(x)$, the slope of the channel midline and the variation of the channel height both affect the effective diffusivity as follows[43]:

$$D(x) = D\left(1 - \left(\frac{\partial z_0(x)}{\partial x}\right)^2 - \frac{1}{12}\left(\frac{\partial z(x)}{\partial x}\right)^2\right) \qquad (15)$$

where $z_0(x) = \frac{z_1(x) + z_2(x)}{2}$ represents the midline of the channel.

To avoid numerical instability in case of large corrections leading to negative D, we use the following equivalent approximation (for small enough $\frac{\partial z(x)}{\partial x}$ and $\frac{\partial z_0(x)}{\partial x}$):

$$D(x) = D\left(\frac{1}{1 + \left(\frac{\partial z_0(x)}{\partial x}\right)^2 + \frac{1}{12}\left(\frac{\partial z(x)}{\partial x}\right)^2}\right) \qquad (16)$$

In the case of localised high $\frac{\partial z(x)}{\partial x}$ and $\frac{\partial z_0(x)}{\partial x}$, this approximation may not reproduce well the fundamental 3D solution in the corresponding localised region of the domain, but the comparisons of Fig. 4 show that such potential cases do not influence much the overall qualitative features of the solution.

We then extend this equation to 2D by separating the two directions (x and y) in a diffusion matrix

$$D(x, y) = ( D(x) 0 0 D(y) ) \qquad (17)$$

Obviously, the use of a diagonal diffusion matrix implies that anisotropies due to local variations in z are accounted for only in the directions aligned with the coordinates (X,Y). A full formal derivation in 3D would identify if ignoring off-diagonal elements is reasonable for a generic geometry. Our numerical simulations suggest that this approximation does not translate into large errors as it produces patterns very similar to those obtained with bona fide 3D models (Fig. 4B).

**Pattern length scale and pattern border curvature in RD simulations under growth.** Lizards grow much more slowly than the time scale required for local RD equilibrium to be achieved. Hence, starting form a mesh of dimensions $L(0) = (L_x(0), L_y(0))$, we simulate isotropic growth by iteratively incrementing the simulation domain size every time the RD steady state is reached (i.e., when the sum of variations of all three components u, v, and w is $<10^{-7}$; see above). More precisely, the simulation domain size at each growth time step τ was updated to $L(\tau) = (1 + \tau\Delta\tau)(L_x(0), L_y(0))$, while the number of simulation elements was kept constant. In Fig. 5, the initial domain size is $L(0) = (1044, 281)$ with 1800 elements in the X-direction. As the Bradley approach[43] of dimensionality-reduction was used in these simulations, $z(x, y)$ and $D(x, y)$—i.e., the position-dependent height and effective diffusion coefficient, respectively—are also updated at each step. The pattern of a dorsal skin patch, with 30% of black scales, from an 18-week-old animal was used as initial condition.

The length scale of real patterns was computed as the mean local width (in number of skin scales) of green labyrinthine stripes (Supplementary Fig. 4). First, 2D images of skin patches of interest were converted to grey-scale and blurred to reduce high-frequency noise (Supplementary Fig. 4A, B). The distribution of pixels grey-scale intensities was then divided in two groups using the k-means clustering algorithm (Supplementary Fig. 4C) to generate a binary image of black and white pixels (Supplementary Fig. 4D). Closed curves connecting white pixels adjacent to black pixels on the binary image were considered as borders of green chains of scales on the corresponding original image (Supplementary Fig. 4E). The distance of each green pixel (i.e., belonging to the cluster with lower grey-scale intensity; Supplementary Fig. 4C) to the closest border was then normalised by half the average distance between the centres of neighbouring skin scales on the corresponding skin patch (i.e., the average skin-scale radius $r_{avg}$). For each pixel belonging to a green stripe, the corresponding normalised distance value is displayed using a colour scale code (Supplementary Fig. 4F). Finally, the global length scale of the pattern was computed as the mean among normalised distances >1. For computing the length scale of simulated patterns, the same procedure was followed; however, as simulation images do not require noise reduction, the clustering threshold was simply set to a value of 127.5, i.e., the centre of the interval of possible pixel intensities.

The average curvature of the pattern border, sampled every $r_{avg}$ along the border (white dots in the inset of Supplementary Fig. 5), is computed using the Menger curvature ($1/r_M$), i.e., the reciprocal of the radius of the circle (yellow, inset of Supplementary Fig. 5) that passes through the current computation point (purple, inset of Supplementary Fig. 5) and two neighbouring points (blue, inset of Supplementary Fig. 5) whose distance to the computation point is equal to $5 \cdot r_{avg}$. To scale out image resolution effects, the radius $r_M$ of the resulting circle is normalised by $r_{avg}$.

**Sensitivity analysis.** The prism colour homogeneity error quantifies how much a generated steady state deviates from a pattern in which each prism has homogeneous colour. This measure was constructed by taking all simulation grid points $i$ that fall inside a prism $p$ and calculating the corresponding prism colour homogeneity error as:

$$e(p) = \min(1 - \text{green}(p), \text{green}(p)), \qquad (18)$$

where $\text{green}(p) = \frac{\sum_{i \in p} g(i)}{|p|}$ and $g(i) \in [0, 1]$ is the green colour intensity of a grid point $i$ (see above). Finally, we compute the pattern colour homogeneity error consisting of $n$ prisms as $\frac{\sum_p e(p)}{n}$ (Supplementary Fig. 1).

**Reporting Summary**. Further information on research design is available in the Nature Research Reporting Summary linked to this article.

## Data availability

The data generated or analysed during this study are included in this published article and its supplementary information file. Very large files with lizard 3D geometries are available from the corresponding author (M.C.M.) on reasonable request.

## Code availability

The GPU-based finite-difference implementation of the RD process on a regular network of spacing ε with periodic or no-flux boundary conditions is provided at https://github.com/LANEvol/RD-3D.git for repeating the numerical simulations presented here.

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

## Acknowledgements

We thank Adrien Debry and Florent Montange for technical assistance with animals, Grigorii Timin, Athanasia Tzika and Carine Langrez for assistance with HREM, and

Antonio Martins for advises on R²OBBIE scans. We thank Szabolcs Zakany and Ebrahim Jahanbakhsh for commenting on the manuscript. This work was supported by grants to MCM from the Georges & Antoine CLARAZ foundation, the Swiss National Science Foundation (FNSNF, grants 31003A_179431 and CR32I3_162743), the International Human Frontier Science Programme Organisation (HFSP RGP0019/ 2017), and the European Research Council (ERC, Advanced grant EVOMORPHYS) under the European Union's Horizon 2020 research and innovation programme. The funding bodies played no role in the design of the study, collection, analysis, and interpretation of data and in writing the manuscript.

## Author contributions

M.C.M. conceived and supervised all aspects of the study. A.F. performed all coding and numerical simulations. A.F. and M.C.M analysed the data. M.C.M wrote the manuscript.

## Competing interests

The authors declare no competing interests.
