## [Peer Review File · Nature Communications]

Reviewer #1 (Remarks to the Author):

Fofonjka and Milinkovitch address a conundrum of pattern formation: how is the Turing picture of pattern formation by a reaction-diffusion system consistent with the sharp colour gradients between the monochromatic scales of e.g. ocellated lizards? They show that the three-dimensional structure of the lizard scales reduces the effective two-dimensional diffusion constants near the periphery of the scales to produce the monochromatic scale pattern observed. This paper is written as a follow-up to an earlier study from the same group (Manukyan et al., 2017; M2017 hereinafter), in which such a reduction was already shown to produce the observed monochromatic scale pattern. This earlier work already suggested that this reduction results from the three-dimensional structure of the scales, but did not show that this structure is sufficient to produce this reduction of the diffusion constants.

I enjoyed reading the paper and I think the results are important for the field of pattern formation. While the analysis is, as far as I can judge, sound, I am not convinced that the manuscript as it stands makes a very good case for its importance: I feel that the structure of the paper, constructed as a follow-up to M2017, with a long introduction of almost equal length to the results section and that provides a blow-by-blow account of this earlier work, is infelicitous. Perhaps it would be preferable to just introduce the basic problem in the introduction, with a reference to the solution suggested by M2017, and to summarise other findings of M2017 when (and if) required (for instance, are the details of the cellular automaton conclusions of M2017 necessary for understanding the results of this paper?).

Perhaps more importantly, I found myself asking: how novel are the results of this paper? M2017 showed that reducing the diffusion constants as a proxy for the thinning of the scales near their boundary is sufficient to produce monochromatic scales; using e.g. Bradley's theory, one should be able to estimate that the reduction of the diffusion constants due to the geometry is sufficient. Therefore, while I agree that one of the authors' achievements here is to make the connection between these different results, I suppose my question is partly: what do we then learn from the 3D simulations (or indeed the reduced 2D simulations)? What we want to learn is, I think, that this geometric effect indeed underpins the monochromatic scales of the ocellated lizard, but the authors cannot quite claim this, because of what I suppose is the usual problem of RD pattern models: does the simplified model used indeed correspond to the real RD process that is going on in the system? For instance, should one expect the same equations (with the same parameter values) to describe pattern formation in zebrafish and ocellated lizards, and what is the long-range factor w molecularly? (It might be appropriate to mention at the top of page 13 that the parameter values used are those previously used for zebrafish by Nakamasu et al. 2009.) I am not expecting the authors to solve this problem, and indeed, they mitigate this issue in different ways: (1) they show that the geometric effect also arises for very different RD models - but is the threshold for monochromaticity also sufficiently low in the unknown, actual RD system describing the scales and, in a given RD model, how does this threshold depend on the parameter choices?; (2) they quantitatively reproduce the pattern at different ages by "growing the animal numerically" and starting the simulation from an observed pattern. This is a beautiful result, but I think the authors could strengthen it by not only comparing the one statistic that is the mean relative length scale of the pattern. However, looking for example at panel B of Fig. 5, it seems that the observed pattern has a finer structure than the simulated one, suggesting that the RD model or the parameter choices are not quite right. I do think it would be useful to discuss these limitations. The authors could perhaps also provide further evidence for this geometric effect actually underpinning

monochromaticity by excluding some alternative hypotheses (e.g. existence of scale substructures that could affect the diffusion constants locally).

Finally, I have a couple of questions about the reduction of the three-dimensional problem to two-dimensional models that I think the authors should address in a revised submission:

(a) On page 16, the authors assume that c does not vary in the z -direction, but it seems easy to prove this assumption from the thinness of the layer: similarly to the derivation of e.g. lubrication theory from the Stokes equations for a viscous fluid, the leading-order term in the RD system is $\partial^2 c / \partial z^2 = 0$, so $c(x, y, z, t) = a(x, y, t)z + b(x, y, t)$. The no-flux conditions on the surfaces are $\partial c / \partial z = 0$, so $a(x, y, t) = 0$, i.e. c is independent of z , and so $Q = c(x, y, t)z(x, y)$.

(b) The RD equation then reduces to $\partial(Q/z) / \partial t = f(Q/z) + D \partial^2(Q/z) / \partial x^2 + D \partial^2(Q/z) / \partial y^2$. The hand-wavy argument above (I suppose that the derivation of the Bradley model, with which I am not familiar, goes through these approximations more carefully) assumes that z varies slowly with x, y , so multiplying through by z , the factor of z can be taken into one derivative to yield the Fick-Jacobs equation. How novel is the authors' derivation? (E.g. has the Fick-Jacobs equation been derived previously for this kind of averaged problem? If so, how is it derived in previous work?)

(c) To avoid negative D in Bradley's model, the authors replace Bradley's expression for D with an expressions that is equivalent in the limit of small $\partial z / \partial x$ and small $\partial z_0 / \partial x$, but I would assume that Bradley's analysis does not apply when D becomes negative (i.e. when $\partial z / \partial x$ and $\partial z_0 / \partial x$ are not small) - perhaps the authors could comment on this in the manuscript.

(d) Moreover, Bradley's analysis reduces a 2D diffusion problem to a 1D problem. The authors extend this to reducing a 3D diffusion problem to a 2D problem by simply introducing a diagonal diffusion matrix. Is it obvious that the Bradley's calculations carry over to the 3D problem in this way? (Why can the diffusion constant D_{xx} not depend on the y -derivatives too? Why can there be no off-diagonal terms in the effective diffusion matrix?)

Minor comments:

(1) A lot of axis labels or figure labels are quite small, and in some cases (axis labels in panel C and colour bar in panel F of Supplementary Figure 2) too small. Increasing the font sizes would make the plots more readable.

(2) In the Materials & Methods subsection "3D reaction-diffusion numerical model", it might be appropriate to replace "using the following scheme (Ref. 37)" with "based on a previously published numerical scheme (Ref. 37)". Also, there should be perhaps be a reference to Ref. 37 in the main text, when mentioning the "3D GPU-based finite-difference numerical approach".

(3) When discussing effective two-dimensional models on page 9, the manuscript refers the reader to the Materials & Methods section, and then suddenly mentions the Fick-Jacobs and Bradley models (with extraneous apostrophes). It would be clearer to introduce the two models (and a brief description as in "with or without an additional position-dependent effective diffusion coefficient") earlier in that paragraph.

(4) The authors switch back and forth between the present and past tense in the results section ("First, we simulate ...", "Second, we tested ..."). A consistent choice of tense would be preferable. (Also, do the paragraphs need to be numbered?)

(5) The manuscript is generally well written, and I found few typos, but one thing I did notice is that the authors use the singular "dynamic" where the plural "dynamics" would be more idiomatic.

Reviewer #2 (Remarks to the Author):

This manuscript is devoted to corroborate the working hypotheses that skin patterns in lizards lead to a discrete cellular automaton, as a kind of numerical discretization of a reaction-diffusion system. Their main explanation of this fact is based on simulations of the RD system in 2D and 3D in a thin slab by some kind of ad hoc reduction. The model is based on the fact that lizards have scales and they are supposed to affect the diffusion coefficient of the different kind of cells originating the colours. However, all of this is based on the assumption that these patterns follow a Turing instability formation. I have two main criticisms to this work:

1. The researchers refer to Kondo's lab work on this respect particularly in fish patterning. However, even Kondo himself and other researchers lately have examined the possibility that the patterns are generated by moving cells due to nonlocal interactions together with reaction terms. This essentially substitutes diffusion by nonlocal interactions. The series of works in this direction are:

Modelling stripe formation in zebrafish: an agent-based approach.
Journal of the Royal Society Interface 12 (112): 2015.

Iridophores as a source of robustness in zebrafish stripes and variability in Danio patterns. Nature Communications 9 (3231): 2018.

(see the references to Kondo's work in these papers) and for the continuum models one can check

O. Trush, C. Liu, X. Han, Y. Nakai, R. Takayama, H. Murakawa, J. A. Carrillo, H. Takechi, S. Hakeda-Suzuki, T. Suzuki, M. Sato, N-cadherin orchestrates self-organization of neurons within a columnar unit in the Drosophila medulla, J. Neuroscience 39, 5861-5880, 2019.

J. A. Carrillo, H. Murakawa, M. Sato, H. Togashi, O. Trush, A population dynamics model of cell-cell adhesion incorporating population pressure and density saturation, J. Theor. Biology 474, 14-24, 2019.

It would be nice some comments of the authors related to this other possible mechanism of pattern formation rather than the Turing mechanism.

2. The details given about the choice of parameters in the RD system are not convincing. Can you explain how the parameters in the reaction terms are chosen? Having a large number of parameters in a RD system is certainly advantageous to be able to match many different pattern formations but this does not mean that the biological principles behind are correct. The model resembles the lizard skin patterns and therefore, the cellular automation associated to it, but this does not give any insight about the biology processes involved. Can the authors comment on this second main

criticism?

The final question I have is based on the second main criticism, how can you be so sure that the Turing mechanism is a good model for pattern formation in the skin of lizards? Can you give a more conclusive answer than the results resemble each other by making a good choice of the parameters?

Reviewer #3 (Remarks to the Author):

In this manuscript, Fofonjka and Milinkovitch model lizard skin patterning using simulations in realistic 3D geometries. This work strongly builds on their previous paper (Manukyan et al. Nature 2017), in which the authors had shown that constrictions implemented by skin scales transform a continuous reaction-diffusion (RD) system into a cellular automaton (CA). The previous model was based on the untested assumption that constrictions locally perturb the movement of patterning molecules, and here the authors set out to test the plausibility of this assumption with rigorous measurements of lizard skin tissue geometry and GPU-accelerated numerical simulations. It might seem like a triviality to test the intuitive assumption of the 2017 Nature paper; however, intuition often fails - in particular for complex systems - and this paper therefore provides the crucial test for the tenets of the influential lizard patterning model. Additionally, the GPU-accelerated RD calculations provide a powerful tool that will be useful for the community and the wider field. The paper is very well written and provides a good introduction to the concepts as well as a compelling narrative. As outlined below, a few questions need to be addressed to bolster the authors' conclusions before this excellent work can be published.

MAJOR POINTS

1) 3D simulations in prisms and lizard scale geometries

a) The authors determined the 3D geometries of representative lizard scales and then set out to perform simulations in realistic domains derived from these measurements. The approach to define regions occupied by melanophores using the polynomial fit to the 3-std interval appears to homogenize the differences that naturally exist in black and green cells, and it seems to overestimate the space occupied by pigment cells on average. It would therefore be more compelling if the authors could perform the simulations in geometries that contain the information from individual scales with non-averaged melanophore occupancies. Can the identities of green and black cells conveyed by u and v concentrations be recovered based on those geometry differences?

b) The authors convincingly show that their findings hold in a variety of RD systems with and without saturation (based on upper and lower concentration limits) and diverse diffusion coefficient ratios from 2 to 10 and 40. Since all simulations are based on dimensionless systems, the authors should calculate the wavelengths of the patterns based on the dispersion relation, so that the readers can relate the diffusion coefficients and spatial pattern scales to domain sizes.

c) The pattern in Fig. 2h is based on the pattern in Fig. 2g as the initial condition. What happens if the system starts from homogeneous initial conditions instead? I suggest to add this simulation as a supplementary figure.

d) On p.7 line 18 and in Fig. 2e, please explain where the value of 0.3 originated, i.e. please show the data that has led to this conclusion.

e) If possible, it would be useful to include an explanation for the readers how the factors that lead to CA behavior from the continuous RD process are quantitatively related, i.e. the diffusion coefficient reduction to $1 - P > 0.8$ in Manukyan et al. 2017, the finding that $h_p/h < 0.3$ in the

present study, as well as the overall diffusion coefficient reduction with the Bradley dimensionality reduction approach.

f) The exact patterning mechanisms in lizards are currently unknown, and in Manukyan et al. Nature 2017, the authors stated that “[...] ‘diffusion’ represents cell movements and the production of long-range cellular projections”. The constrictions simulated in the current paper would certainly affect molecule and cell movements. However, the situation might be different for long-range cellular projections, which could in principle simply grow under or around the constrictions. The authors should therefore discuss potential limitations of their model for this kind of mechanism in the paper. If technically feasible, it would be ideal to base this discussion on simulations with long-range cellular projections (e.g. PMID 32579554).

2) Dimensionality reduction

a) Fig. 2h and Fig. 2j clearly show that there are inhomogeneities in the z direction. This is in contrast to the statement on p. 16 line 19 “[...] we assume that there is no variation of the 3D density c in the z direction”. Please explain in the manuscript to what extent this simplification affects the conclusions.

b) In Fig. 4, please show the initial conditions.

c) In Fig. 4, please show the cross section for u and v concentrations (not only the thresholded green values), so that the readers can better evaluate similarities or differences between the different approaches.

3) Growth simulations and spatial scales

a) I suggest to change the title of the paper (remove “On Growth and Form:” and “Growing”) since growth is only a minor aspect and not the primary focus of the paper.

b) The paper is very vague about how growth was modeled. Please provide more technical details in the Methods section. For example, please explain how the elements were split after each growth step, whether the concentrations of the reactants were kept constant after splitting the elements (which might be incompatible with mass conservation considerations), whether growth was isotropic, how many elements of what size the final domain comprised, etc.

c) Please give a reference or show data for the statement on p. 18 that “Lizards grow much more slowly than the time scale required for local RD equilibrium to be achieved”.

d) Please add the initial condition and the growth curve for the simulations mentioned on p. 10 line 20, so that the readers can better evaluate the data.

e) In Fig. 5a, the match between experiments and simulations is convincing. However, it is surprising that the spatial patterns of the simulations do not look more like the observed patterns, given that real animals appear to preserve a rudimentary organization of the initial pattern (Fig. 1b-d, Fig. 5b-c) and the simulations were started with the real animal pattern as the initial condition. Please mention this caveat in the paper and discuss potential implications for how well the model captures the real patterning process.

f) Growth in Supplementary Movie 1 is not apparent, and it would be better to show non-normalized data.

g) Please also show non-normalized images in Fig. 5, so that the pattern relative to the growing domain can be evaluated.

h) In Fig. 5b-c, please show identical regions from the simulations (Supplementary Movie 1 indicates that different regions were chosen in panels b and c). Please also adjust Supplementary Movie 1 accordingly.

i) In general, all components in RD networks have the same range since there is only a single wavelength in such systems. The experimental and simulated findings in Fig. 5 that green scales get

confined to a single tier over time, whereas the spacing of black scales stays wider, are therefore surprising. Please clarify in the paper whether the range of the reactants u and v also changes or whether the differences between green and black scales are due to the arbitrary “clamping” of green identity through the function on p. 13 line 6.

j) It would be very convincing if the authors could use their model to make new predictions rather than purely reproducing existing data. The ideal test would be to experimentally induce or remove constrictions in lizard scales and observe pattern defects, although this is most likely not feasible and clearly beyond the scope of the study. But it would already be illuminating if the authors could model what happens in a spatially inhomogeneous field with differently sized scales. In the best case, these simulations could be compared to spontaneous defects potentially observed in a subset of the authors’ lizards, and in the worst case the simulations would provide a prediction that could be tested in the future. Similarly, does the new simulation approach still yield patterned scales like the ones observed on the tail as shown in Manukyan et al. 2017?

MINOR POINTS

1) References and formatting

- a) On p. 3 line 35, it would be good to include more recent modeling approaches that take into account the important roles of iridophores in agent-based models as well as their continuum limit (e.g. PMIDs 26538560 and 30104716).
- b) On p.9 line 8, it would be better to invert the “1D-to-2D” to make it similar to the wording “3D-to-2D” in the line above.
- c) Please explain the “T” in the legend for Fig. 2a,d.

2) Methods description

- a) On p. 13 line 3, the superfluous “ $D_u = 13.5$ ” should be removed.
- b) $z^{\text{skin_top}}$ on p.15 does not seem to be defined. Please clarify.

3) Simulations and software

- a) It would be useful to give a rough idea to the readers how long the simulations take on an average computer, such that the speed-up benefit on p. 9 line 28 can be appreciated (e.g. does the 10-20 time gain correspond to 10 seconds, 10 hours, or 10 days?). This information can currently only be found in the software package, and the README file indicates that the gain will most likely be in the seconds to minutes range.
- b) Please add the red dot at the origin in Fig. 5a, just like in Supplementary Movie 1.
- c) To make the software widely applicable, it would be useful to add the other network types to the software as well (e.g. /gaussian_bumps/*.txt etc.) and to extend the README file with other input options for the simulations, so that 2D projection simulations etc. can also easily be executed.

Research Article: NCOMMS-20-25609: 'Reaction-diffusion in a Growing 3D Domain of Skin Scales Generates a Discrete Cellular Automaton' by Fofonjka & Milinkovitch.

ANSWERS TO REVIEWER COMMENTS

Reviewer #1 (Remarks to the Author):

Fofonjka and Milinkovitch address a conundrum of pattern formation: how is the Turing picture of pattern formation by a reaction-diffusion system consistent with the sharp colour gradients between the monochromatic scales of e.g. ocellated lizards? They show that the three-dimensional structure of the lizard scales reduces the effective two-dimensional diffusion constants near the periphery of the scales to produce the monochromatic scale pattern observed. This paper is written as a follow-up to an earlier study from the same group (Manukyan et al., 2017; M2017 hereinafter), in which such a reduction was already shown to produce the observed monochromatic scale pattern. This earlier work already suggested that this reduction results from the three-dimensional structure of the scales, but did not show that this structure is sufficient to produce this reduction of the diffusion constants.

I enjoyed reading the paper and I think the results are important for the field of pattern formation.

Answer: we thank the Reviewer for her/his positive comments, they allowed us to substantially improve the manuscript.

Please, note that, in comparison to the first submitted version:

- The Discussion section is entirely new.
- There are 4 new Supplementary Figures (Suppl. Figs 1, 2, 5, and 6). The old Supplementary Figures 1 and 2 become Suppl. Figs 3 and 4.
- Supplementary Video 1 has been modified to show the domain growth.
- All other changes in the text are marked.

While the analysis is, as far as I can judge, sound, I am not convinced that the manuscript as it stands makes a very good case for its importance: I feel that the structure of the paper, constructed as a follow-up to M2017, with a long introduction of almost equal length to the results section and that provides a blow-by-blow account of this earlier work, is infelicitous. Perhaps it would be preferable to just introduce the basic problem in the introduction, with a reference to the solution suggested by M2017, and to summarise other findings of M2017 when (and if) required (for instance, are the details of the cellular automaton conclusions of M2017 necessary for understanding the results of this paper?).

Answer: we think it is a service to the reader to (i) introduce the suggestion (made in the M2017 paper) that geometry transforms a microscopic reaction diffusion system into a meso/macrosopic cellular automaton as well as (ii) explain the fact that the M2017 paper does not demonstrate that the actual geometry of the lizard scales is sufficient to bring about this phenomenon. In my experience, some people are confused if the biological *versus* physical and modelling concepts are not sufficiently articulated in the introduction. We anyway followed the advise of the reviewer and substantially reduced the length of the description of the M2017 paper (by entirely skipping the previous third point about the slow down, over time, of the scale switching process) while keeping the last three paragraphs of the introduction. These paragraphs are essential to the present manuscript as they explain (i) what was the untested assumption of M2017 paper, (ii) how we test this assumption with 3D simulations (despite homogeneous diffusion coefficients), (iii) that the scale-by-scale snapping process is robust to variation of the RD model, (iv) that real scale geometry (and not only idealised ones) do produce the

same phenomenon, and (v) that growth affects both the relative length scale of the pattern and the curvature of the pattern motifs (the latter was not included in the first version of the manuscript and is now introduced following another comment of the same reviewer; see below).

Perhaps more importantly, I found myself asking: how novel are the results of this paper? M2017 showed that reducing the diffusion constants as a proxy for the thinning of the scales near their boundary is sufficient to produce monochromatic scales; using e.g. Bradley's theory, one should be able to estimate that the reduction of the diffusion constants due to the geometry is sufficient. Therefore, while I agree that one of the authors' achievements here is to make the connection between these different results, I suppose my question is partly: what do we then learn from the 3D simulations (or indeed the reduced 2D simulations)? What we want to learn is, I think, that this geometric effect indeed underpins the monochromatic scales of the ocellated lizard, but the authors cannot quite claim this, because of what I suppose is the usual problem of RD pattern models: does the simplified model used indeed correspond to the real RD process that is going on in the system? For instance, should one expect the same equations (with the same parameter values) to describe pattern formation in zebrafish and ocellated lizards, and what is the long-range factor w molecularly? (It might be appropriate to mention at the top of page 13 that the parameter values used are those previously used for zebrafish by Nakamasu et al. 2009) I am not expecting the authors to solve this problem, ...

Answer: we appreciate the interest of the Reviewer for a biologically-accurate description of the self-organisational patterning system at work in zebrafish vs. ocellated lizard, and we are glad that she/he realises that this is probably not achievable in the ocellated lizard given that this model is very far from being as amenable to cell-biology investigations as the zebrafish. Note also that the two biological systems (zebrafish and lizard), although obviously related, would likely require substantially-different agent-based models. Indeed, it is likely that the actual patterning processes of the two species differ by at least some aspects ... given the minimum of 2 x 350 million years separating the two species. This is the very reason why we favour a more general (Turing reaction diffusion, RD) model rather than, for example, a more complete agent-based model integrating all what is known of the cell biology of zebrafish chromatophores. Instead, our paper specifically focuses on the question whether periodic thinning of the skin is sufficient to transform a continuous RD (cRD) system into a CA-like discrete system, a question that has been suggested, but not formally answered, in the M2017 paper. As noted by Reviewer 3:

"It might seem like a triviality to test the intuitive assumption of the 2017 Nature paper; however, intuition often fails - in particular for complex systems - and this paper therefore provides the crucial test for the tenets of the influential lizard patterning model."

Despite that they can be less complete than other models, RD systems are likely to be sufficiently general to capture the effect of geometry on the patterning. We would love to see the question investigated with a 3D agent-based model (so far, the published studies on skin colour patterning are in 2D), but that's beyond the scope of our study.

In the revised version of our manuscript, we now add most of these considerations (hence, the references to recent agent-based studies) in the entirely new Discussion section. We particularly make clear that we investigate the concept of geometry transforming cRD into CA and do not attempt to precisely model all aspects of skin colour patterning. The agent-based models are also now mentioned in the introduction.

and indeed, they mitigate this issue in different ways: (1) they show that the geometric effect also arises for very different RD models - but is the threshold for monochromaticity also sufficiently low in the unknown, actual RD system describing the scales and, in a given RD model, how does this threshold depend on the parameter choices?;

Answer: We indeed strongly mitigate the issue of model-dependence of our results by using vastly different RD models (note that they don't merely differ by parameter values). Surely, the observation of a scale-by-scale colour patterning using the model from Nakamasu et al. 2009, but also with two different two-component models, is a strong indication that the effect of scale geometry on the discretisation of the pattern is somewhat general. It does not demonstrate that the same effect will necessarily be exhibited in more complete agent-based models (until this is tested), but it strongly suggests it would. Again, we now introduce these considerations in the Discussion section.

(2) they quantitatively reproduce the pattern at different ages by "growing the animal numerically" and starting the simulation from an observed pattern. This is a beautiful result, but I think the authors could strengthen it by not only comparing the one statistic that is the mean relative length scale of the pattern. However, looking for example at panel B of Fig. 5, it seems that the observed pattern has a finer structure than the simulated one, suggesting that the RD model or the parameter choices are not quite right. I do think it would be useful to discuss these limitations.

Answer: We thank the reviewer for his appreciation of our numerical growth analysis. We now follow the suggestion of the Reviewer by supplementing the length scale analysis with an additional statistics that captures the transformation of the early pattern (large bands) into the mature labyrinthine (convoluted finer bands).

The authors could perhaps also provide further evidence for this geometric effect actually underpinning monochromaticity by excluding some alternative hypotheses (e.g. existence of scale substructures that could affect the diffusion constants locally).

Answer: We now indicate in the Discussion section that our histological analyses do not evidence any scale substructure (beyond the thinning of the skin, *per se*) that could affect the diffusion constants locally.

Finally, I have a couple of questions about the reduction of the three-dimensional problem to two-dimensional models that I think the authors should address in a revised submission:

(a) On page 16, the authors assume that c does not vary in the z -direction, but it seems easy to prove this assumption from the thinness of the layer: similarly to the derivation of e.g. lubrication theory from the Stokes equations for a viscous fluid, the leading-order term in the RD system is $\partial^2 c / \partial z^2 = 0$, so $c(x,y,z,t) = a(x,y,t)z + b(x,y,t)$. The no-flux conditions on the surfaces are $\partial c / \partial z = 0$, so $a(x,y,t) = 0$, i.e. c is independent of z , and so $Q = c(x,y,t)z(x,y)$.

(b) The RD equation then reduces to $\partial(Q/z)/\partial t = f(Q/z) + D\partial^2(Q/z)/\partial x^2 + D\partial^2(Q/z)/\partial y^2$. The hand-wavy argument above (I suppose that the derivation of the Bradley model, with which I am not familiar, goes through these approximations more carefully) assumes that z varies slowly with x,y , so multiplying through by z , the factor of z can be taken into one derivative to yield the Fick-Jacobs equation. How novel is the authors' derivation? (E.g. has the Fick-Jacobs equation been derived previously for this kind of averaged problem? If so, how is it derived in previous work?)

Answer: The reviewer is right. When performing formal scale analysis, one can justify that $\partial c / \partial z = 0$ because the length scale of the pattern is larger than the thickness of the domain. We now indicate exactly that point in the revised version of the manuscript.

First we indicate in the Materials and Methods that this is a standard procedure. Second, we indicate that *“the assumption of no variation of the density c in the z direction is jointly justified by (i) the thickness of the domain in z being smaller than the typical length-scale of the pattern, (ii) the variation in z of u , v , and w at any (x,y) point being much smaller than the difference of these concentrations between green and black scales, (as shown in bona fide 3D simulations; Fig. 2F) and (iii) the no-flux boundary condition of the domain.”*

(c) To avoid negative D in Bradley's model, the authors replace Bradley's expression for D with an expressions that is equivalent in the limit of small $\partial z/\partial x$ and small $\partial z_0/\partial x$, but I would assume that Bradley's analysis does not apply when D becomes negative (i.e. when $\partial z/\partial x$ and $\partial z_0/\partial x$ are not small) - perhaps the authors could comment on this in the manuscript.

Answer: The Reviewer is correct, if D becomes negative, the equivalent approximation that we use avoids numerical instability. We now indicate:

“In the case of localised low $D(x)$, this approximation may not reproduce well the fundamental 3D solution in the corresponding localised region, but the comparisons of Fig. 4 show that such potential cases do not influence much the overall qualitative features of the solution”.

(d) Moreover, Bradley's analysis reduces a 2D diffusion problem to a 1D problem. The authors extend this to reducing a 3D diffusion problem to a 2D problem by simply introducing a diagonal diffusion matrix. Is it obvious that the Bradley's calculations carry over to the 3D problem in this way? (Why can the diffusion constant D_{xx} not depend on the y -derivatives too? Why can there be no off-diagonal terms in the effective diffusion matrix?)

Answer: The Reviewer is correct that the diagonal matrix is an approximation. We now indicate that *“Obviously, the use of a diagonal diffusion matrix implies that anisotropies due to local variations in z are accounted for only in the directions aligned with the coordinates (X,Y) . A full formal derivation in 3D would identify if ignoring off-diagonal elements is reasonable for a generic geometry. Our numerical simulations suggest that this approximation does not translate into large errors as it produces patterns very similar to those obtained with bona-fide 3D models (Fig. 4B).”*

Minor comments:

(1) A lot of axis labels or figure labels are quite small, and in some cases (axis labels in panel C and colour bar in panel F of Supplementary Figure 2) too small. Increasing the font sizes would make the plots more readable.

Answer: We corrected the problem in the figures mentioned by the Reviewer and we increased the font size in other figures.

(2) In the Materials & Methods subsection "3D reaction-diffusion numerical model", it might be appropriate to replace "using the following scheme (Ref. 37)" with "based on a previously published numerical scheme (Ref. 37)". Also, there should be perhaps be a reference to Ref. 37 in the main text, when mentioning the "3D GPU-based finite-difference numerical approach".

Answer: These changes have been introduced.

(3) When discussing effective two-dimensional models on page 9, the manuscript refers the reader to the Materials & Methods section, and then suddenly mentions the Fick-Jacobs and Bradley models (with extraneous apostrophes). It would be clearer to

introduce the two models (and a brief description as in "with or without an additional position-dependent effective diffusion coefficient") earlier in that paragraph.

Answer: These changes have been introduced.

(4) The authors switch back and forth between the present and past tense in the results section ("First, we simulate ...", "Second, we tested ..."). A consistent choice of tense would be preferable. (Also, do the paragraphs need to be numbered?)

Answer: Present tense is now used throughout.

(5) The manuscript is generally well written, and I found few typos, but one thing I did notice is that the authors use the singular "dynamic" where the plural "dynamics" would be more idiomatic.

Answer: We now use "dynamics".

Reviewer #2 (Remarks to the Author):

This manuscript is devoted to corroborate the working hypotheses that skin patterns in lizards lead to a discrete cellular automaton, as a kind of numerical discretization of a reaction-diffusion system. Their main explanation of this fact is based on simulations of the RD system in 2D and 3D in a thin slab by some kind of adhoc reduction. The model is based on the fact that lizards have scales and they are supposed to affect the diffusion coefficient of the different kind of cells originating the colours. However, all of this is based on the assumption that these patterns follow a Turing instability formation. I have two main criticisms to this work:

Answer: We thank the reviewer for these constructive comments, allowing us to substantially improve the manuscript.

Please, note that, in comparison to the first submitted version:

- The Discussion section is entirely new.
- There are 4 new Supplementary Figures (Suppl. Figs 1, 2, 5, and 6). The old Supplementary Figures 1 and 2 become Suppl. Figs 3 and 4.
- Supplementary Video 1 has been modified to show the domain growth.
- All other changes in the text are marked.

1. The researchers refer to Kondo's lab work on this respect particularly in fish patterning. However, even Kondo himself and other researchers lately have examined the possibility that the patterns are generated by moving cells due to nonlocal interactions together with reaction terms. This essentially substitute diffusion by nonlocal interactions. The series of works in this direction are:

Modelling stripe formation in zebrafish: an agent-based approach.

Journal of the Royal Society Interface 12 (112): 2015.

Iridophores as a source of robustness in zebrafish stripes and variability in Danio patterns. Nature Communications 9 (3231): 2018.

(see the references to Kondo's work in these papers) and for the continuum models one can check

O. Trush, C. Liu, X. Han, Y. Nakai, R. Takayama, H. Murakawa, J. A. Carrillo, H. Takechi, S. Hakeda-Suzuki, T. Suzuki, M. Sato, N-cadherin orchestrates self-organization of neurons within a columnar unit in the Drosophila medulla, J. Neuroscience 39, 5861-5880, 2019.

J. A. Carrillo, H. Murakawa, M. Sato, H. Togashi, O. Trush, A population dynamics model of cell-cell adhesion incorporating population pressure and density saturation, J. Theor. Biology 474, 14-24, 2019.

It would be nice some comments of the authors related to this other possible mechanism of pattern formation rather than the Turing mechanism.

Answer: We appreciate the interest of the Reviewer for a biologically-accurate description of the self-organisational patterning system at work in zebrafish vs. ocellated lizard. Note that this is probably not achievable in the ocellated lizard given that this model is very far from being as amenable to cell-biology investigations as the zebrafish. Note also that the two biological systems (zebrafish and lizard), although obviously related, would likely require substantially-different agent-based models. Indeed, it is likely that the actual patterning processes of the two species differ by at least some aspects ... given the minimum 2 x 350 million years separating the two species. This is the very reason why we favour a more general (Turing reaction diffusion, RD) model rather than, for example, a more complete agent-based model integrating all what is

known of the cell biology of zebrafish chromatophores. Instead, our paper specifically focuses on the question whether periodic thinning of the skin is sufficient to transform a continuous RD (cRD) system into a CA-like discrete system, a question that has been suggested, but not formally answered, in the M2017 paper. As noted by Reviewer 3:

“It might seem like a triviality to test the intuitive assumption of the 2017 Nature paper; however, intuition often fails - in particular for complex systems - and this paper therefore provides the crucial test for the tenets of the influential lizard patterning model.”

Despite that they can be less complete than other models, RD systems are likely to be sufficiently general to capture the effect of geometry on the patterning. We would love to see the question investigated with a 3D agent-based model (so far, the published studies for skin colour patterning are in 2D), but that’s beyond the scope of our study.

We now add these considerations (hence, the references to recent agent-based studies, as suggested by the Reviewer) in the entirely new Discussion section. We particularly make clear that we investigate the concept of geometry transforming cRD into CA and do not attempt to precisely model all aspects of skin colour patterning.

2. The details given about the choice of parameters in the RD system are not convincing. Can you explain how the parameters in the reaction terms are chosen? Having a large number of parameters in a RD system is certainly advantageous to be able to match many different pattern formations but this does not mean that the biological principles behind are correct. The model resembles the lizard skin patterns and therefore, the cellular automation associated to it, but this does not give any insight about the biology processes involved. Can the authors comment on this second main criticism? The final question I have is based on the second main criticism, how can you be so sure that the Turing mechanism is a good model for pattern formation in the skin of lizards? Can you give a more conclusive answer than the results resemble each other by making a good choice of the parameters?

Answer: Again, we don’t aim to precisely model the cellular biological mechanism underpinning the patterning process. We aim to test whether realistic geometry alone can transform a cRD into a discrete CA. And we demonstrate it can. This is far from trivial. Regarding the values of parameters, we simply tune them, starting from the values proposed by Nakamasu et al. 2009. All signs of the reaction-term coefficients are identical to those of Nakamasu et al. 2009; this is important because it reflects activation/inhibition observed in a true biological system (the zebrafish). Their absolute values are also similar. This is now indicated in the Materials and Methods section.

Note that agent-based models also (i) fit the model and its parameters to the expected pattern and dynamics (e.g., Owen et al. eLife 2020) and (ii) incorporate some level of idealisation of the particles that indirectly take into account cell behaviours (e.g., Volkening and Sandstede 2015; 2018). These agent-based models and studies are fascinating and we think that the two approaches (RD and agent-based) are complementary. We now discuss all these points in the manuscript and refer to the agent-based publications both in the introduction and in the Discussion.

Reviewer #3 (Remarks to the Author):

In this manuscript, Fofonjka and Milinkovitch model lizard skin patterning using simulations in realistic 3D geometries. This work strongly builds on their previous paper (Manukyan et al. Nature 2017), in which the authors had shown that constrictions implemented by skin scales transform a continuous reaction-diffusion (RD) system into a cellular automaton (CA). The previous model was based on the untested assumption that constrictions locally perturb the movement of patterning molecules, and here the authors set out to test the plausibility of this assumption with rigorous measurements of lizard skin tissue geometry and GPU-accelerated numerical simulations. It might seem like a triviality to test the intuitive assumption of the 2017 Nature paper; however, intuition often fails - in particular for complex systems - and this paper therefore provides the crucial test for the tenets of the influential lizard patterning model. Additionally, the GPU-accelerated RD calculations provide a powerful tool that will be useful for the community and the wider field. The paper is very well written and provides a good introduction to the concepts as well as a compelling narrative.

Answer: We thank very much the Reviewer for her/his very positive assessment of our work as well as for the extensive constructive comments listed below, allowing us to substantially improve the manuscript.

Please, note that, in comparison to the first submitted version:

- The Discussion section is entirely new.
- There are 4 new Supplementary Figures (Suppl. Figs 1, 2, 5, and 6). The old Supplementary Figures 1 and 2 become Suppl. Figs 3 and 4.
- Supplementary Video 1 has been modified to show the domain growth.
- All other changes in the text are marked.

As outlined below, a few questions need to be addressed to bolster the authors' conclusions before this excellent work can be published.

MAJOR POINTS

1) 3D simulations in prisms and lizard scale geometries

a) The authors determined the 3D geometries of representative lizard scales and then set out to perform simulations in realistic domains derived from these measurements. The approach to define regions occupied by melanophores using the polynomial fit to the 3-std interval appears to homogenize the differences that naturally exist in black and green cells, and it seems to overestimate the space occupied by pigment cells on average. It would therefore be more compelling if the authors could perform the simulations in geometries that contain the information from individual scales with non-averaged melanophore occupancies. Can the identities of green and black cells conveyed by u and v concentrations be recovered based on those geometry differences?

Answer: Yes, indeed, green and black scales can readily be distinguished in histological sections on the basis of the distributions of pigments, especially melanosomes. This is illustrated in Fig. 3A-C. It seems to us very difficult to consider a different 3D domain of green vs. black scale occupancy by chromatophores. Indeed, given that scales change colour from green to black and from black to green, we would need to dynamically and locally change the domain size as the scales change colour. To assess the dynamics of domain size variation, we would need to sample changing scales at various moments of colour flipping. This is not entirely impossible but would require months of sampling. In addition, the distribution of pigments does not equate to the distribution of cells. It is

likely that the massive amount of pigments at the top of the black scales is due to melanophores sending projections at the top of the dermis rather than the cells moving themselves. In other words, we currently do not have data justifying to make the model more complicated by introducing an additional parameter that would define the local thickness of the domain as a function of u and/or v . Hence, although the remark of the Reviewer is well taken, and could bring very interesting theoretical results, we feel that such an analysis is beyond the scope of our manuscript. We however now indicate this possibility as a perspective in the Discussion section.

b) The authors convincingly show that their findings hold in a variety of RD systems with and without saturation (based on upper and lower concentration limits) and diverse diffusion coefficient ratios from 2 to 10 and 40. Since all simulations are based on dimensionless systems, the authors should calculate the wavelengths of the patterns based on the dispersion relation, so that the readers can relate the diffusion coefficients and spatial pattern scales to domain sizes.

Answer: Indeed, we could obtain the wavelengths from linearised computation of the Nakamasu et al., Gray-Scott and Schnackenberg models and compare these wavelengths to the size of hexagons. However, the information is already available in Fig. 2. Indeed, the Turing length-scales correspond to the typical pattern shown in the simulations without scale boundaries (so Fig. 2C for Nakamasu et al., the right panel of Fig. 2G for Gray-Scott and the right panel of Fig. 2I for Schnackenberg). These can be compared with the size of the hexagonal scale, which are visible in the simulations of Fig. 2F, 2H (right panel) and 2J (right panel). We now explain this point in the text.

c) The pattern in Fig. 2h is based on the pattern in Fig. 2g as the initial condition. What happens if the system starts from homogeneous initial conditions instead? I suggest to add this simulation as a supplementary figure.

Answer: We performed new simulations with alternative initial conditions and the results are now illustrated in a new supplementary figure (Supplementary Fig. 2).

d) On p.7 line 18 and in Fig. 2e, please explain where the value of 0.3 originated, i.e. please show the data that has led to this conclusion.

Answer: This is an empirical observation: when comparing the thickness of the skin in inter-scale skin *versus* the middle of a scale, we obtain a ratio of about 0.3. We now indicate this point in the text and also provide a new supplementary figure 1 showing how varying h_p/h affects the-scale-by-scale discretisation.

e) If possible, it would be useful to include an explanation for the readers how the factors that lead to CA behavior from the continuous RD process are quantitatively related, i.e. the diffusion coefficient reduction to $1 - P > 0.8$ in Manukyan et al. 2017, the finding that $h_p/h < 0.3$ in the present study, as well as the overall diffusion coefficient reduction with the Bradley dimensionality reduction approach.

Answer: A quantitative comparison is quite difficult. Indeed, in Manukyan et al. 2017, a sensitivity analysis identified that the CA-like behaviour appears for a value of $P \approx 0.2$ at the border of scales. It is important to note that the thickness of scale borders in that model is minimal (*i.e.*, about one discretisation element). On the other hand, in the 3D simulations performed here, although the scale border is discrete for prisms (T in Fig. 2D), it is continuous for Gaussian bumps as well as for realistic geometries derived from episcopic microscopy. It is very difficult to take into account such variation in the 'width' of interscale skin for performing a quantitative comparison of P and h_p/h . We introduce now in the Discussion section that the progressive thinning of inter-scale skin generates

the same qualitative transitions of the pattern, in both 2D (ref. 27) and 3D simulations (new Supplementary Fig. 1): from unaffected by scale boundaries (right inset in Supplementary Fig. 1) to scale-by-scale, with motifs that are first 2 to 3 scales wide (central inset) then 1 or 2 scales wide (left inset).

f) The exact patterning mechanisms in lizards are currently unknown, and in Manukyan et al. Nature 2017, the authors stated that “[...] ‘diffusion’ represents cell movements and the production of long-range cellular projections”. The constrictions simulated in the current paper would certainly affect molecule and cell movements. However, the situation might be different for long-range cellular projections, which could in principle simply grow under or around the constrictions. The authors should therefore discuss potential limitations of their model for this kind of mechanism in the paper. If technically feasible, it would be ideal to base this discussion on simulations with long-range cellular projections (e.g. PMID 32579554).

Answer: We follow the Reviewer suggestion and now added a paragraph in the Discussion section on why long-range cellular projections would be impeded by the small thickness of the inter-scale skin. We start by discussing the fact that some long-range interactions among chromatophores, clearly involved in skin colour patterning, are mediated by long cell projections (airinemes) and that, more generally, recent in-vivo and numerical analyses (Rosenbauer et al. 2020) suggest cytoneme-mediated morphogen gradient establishment as a fundamental principles in the patterning of rapidly-expanding tissues, such as in early development. We then discuss more directly the question whether the inter-scale constrictions are expected to affect the ‘diffusion’ of long-range cellular projections similarly to free-diffusing morphogens: we indicate that (i) xanthophore blebs are recognised by macrophages that then drag airineme vesicles and filaments (Eom and Parichy 2017) and (ii) that the wandering of the macrophages (hence, of the xanthophores cellular long projections) do not appear to be directional. The latter point suggests that simple diffusion appropriately models long-range interactions and, hence, that depth reduction of the inter-scale skin is likely to affect long-range interactions as if they were caused by diffusing morphogens. All these points are now introduced in the Discussion section.

2) Dimensionality reduction

a) Fig. 2h and Fig. 2j clearly show that there are inhomogeneities in the z direction. This is in contrast to the statement on p. 16 line 19 “[...] we assume that there is no variation of the 3D density c in the z direction”. Please explain in the manuscript to what extent this simplification affects the conclusions.

Answer: We now better explain in the Material and Methods that the assumption of no variation of the density c in the z direction is jointly justified by (i) the thickness of the domain in z being smaller than the typical length-scale of the pattern, (ii) the variation in z of u , v , and w at any (x,y) point being much smaller than the difference of these concentrations between green and black scales, (as shown in bona fide 3D simulations; Fig. 2F) and (iii) the no-flux boundary condition of the domain.

b) In Fig. 4, please show the initial conditions.

c) In Fig. 4, please show the cross section for u and v concentrations (not only the thresholded green values), so that the readers can better evaluate similarities or differences between the different approaches.

Answer: We now include the cross section of u and v in Fig. 4 (new panel C) . The initial condition is the following:

3) Growth simulations and spatial scales

a) I suggest to change the title of the paper (remove “On Growth and Form:” and “Growing”) since growth is only a minor aspect and not the primary focus of the paper.

Answer: We have removed “On Growth and Form:” but prefer to keep the word “Growing” as this is still an important aspect of the paper.

b) The paper is very vague about how growth was modeled. Please provide more technical details in the Methods section. For example, please explain how the elements were split after each growth step, whether the concentrations of the reactants were kept constant after splitting the elements (which might be incompatible with mass conservation considerations), whether growth was isotropic, how many elements of what size the final domain comprised, etc.

c) Please give a reference or show data for the statement on p. 18 that “Lizards grow much more slowly than the time scale required for local RD equilibrium to be achieved”.

d) Please add the initial condition and the growth curve for the simulations mentioned on p. 10 line 20, so that the readers can better evaluate the data.

Answer: These questions are now answered by the extension of the paragraph on how growth was modelled. The animal growth curve is also now provided as an inset of Fig. 5A. From these data, it is now clear that growth takes months/years whereas some scales change colour in days/weeks.

e) In Fig. 5a, the match between experiments and simulations is convincing. However, it is surprising that the spatial patterns of the simulations do not look more like the observed patterns, given that real animals appear to preserve a rudimentary organization of the initial pattern (Fig. 1b-d, Fig. 5b-c) and the simulations were started with the real animal pattern as the initial condition. Please mention this caveat in the paper and discuss potential implications for how well the model captures the real patterning process.

Answer: Note that the initial pattern is fully obliterated in real lizards as well, although possibly more slowly than in the simulations. This is now indicated in the Figure 1 legend. Yet, the Reviewer is correct: although the dynamics of the mean relative length-scale are highly similar in the observed and simulated patterns, the two pattern might differ in other details. We now added in the Discussion section that the predictability of the model could be improved by using more subtle initial conditions, *i.e.*, real colours instead of assigning all green scales to $(u,v,w)=(1.2,6.6,2.3)$ and all black scales to $(u,v,w)=(5.3,0.92,4)$, followed by addition of $\pm 1\%$ random noise.

f) Growth in Supplementary Movie 1 is not apparent, and it would be better to show non-normalized data.

g) Please also show non-normalized images in Fig. 5, so that the pattern relative to the growing domain can be evaluated.

Answer: We modified Supplementary Movie 1 by including normalised data in order to clearly shows how the domain and scales grow. We did not add the non-normalised images in Fig. 5 because it was making the figure very messy and they are now available in the supplementary Movie.

h) In Fig. 5b-c, please show identical regions from the simulations (Supplementary Movie 1 indicates that different regions were chosen in panels b and c). Please also adjust Supplementary Movie 1 accordingly.

Answer: We now show identical regions in Fig. 5 and adjusted the Supplementary Movies accordingly.

i) In general, all components in RD networks have the same range since there is only a single wavelength in such systems. The experimental and simulated findings in Fig. 5 that green scales get confined to a single tier over time, whereas the spacing of black scales stays wider, are therefore surprising. Please clarify in the paper whether the range of the reactants u and v also changes or whether the differences between green and black scales are due to the arbitrary “clamping” of green identity through the function on p. 13 line 6.

Answer: Indeed, there is a single wavelength (corresponding to the combination of the green and black stripes). The relative widths of the green and black stripes simply depend on how u and v are translated into visible green and black colours. In our simulations, greenness is computed with $g = \min(\max((v - u)/(v + u) + 0.5, 0), 1)$ (cf. Material and Methods page 18). This insures that green and black stripes have the same width. There is no reason that the biological process follows such a perfectly symmetrical scheme. As an example, if another value than 0.5 was used in the max relation above, one would generated different widths of green and black stripes. In other words, the biological translation of the activator/inhibitor ‘concentrations’ is unknown and could be substantially different from the way it is computed in the simulations. This is now indicated in the Methods section.

j) It would be very convincing if the authors could use their model to make new predictions rather than purely reproducing existing data. The ideal test would be to experimentally induce or remove constrictions in lizard scales and observe pattern defects, although this is most likely not feasible and clearly beyond the scope of the study. But it would already be illuminating if the authors could model what happens in a spatially inhomogeneous field with differently sized scales. In the best case, these simulations could be compared to spontaneous defects potentially observed in a subset of the authors’ lizards, and in the worst case the simulations would provide a prediction that could be tested in the future. Similarly, does the new simulation approach still yield patterned scales like the ones observed on the tail as shown in Manukyan et al. 2017?

Answer: We are glad the Reviewer realises these kind of (very difficult) lab experiments are beyond the scope of the manuscript. We are actually trying various experiments for perturbing the system ... without much luck for the moment. However, as suggested by the Reviewer, we now performed simulations with tail scales (much larger than body scales) and indeed obtain realistic patterns, i.e. with within-scale patterns. These new results are now incorporated in an additional supplementary Figure 6.

MINOR POINTS

1) References and formatting

a) On p. 3 line 35, it would be good to include more recent modeling approaches that take into account the important roles of iridophores in agent-based models as well as their continuum limit (e.g. PMIDs 26538560 and 30104716).

Answer: These two references (and others) have been added in the revised version of our manuscript (both in the Introduction and Discussion sections).

b) On p.9 line 8, it would be better to invert the “1D-to-2D” to make it similar to the wording “3D-to-2D” in the line above.

Answer: This has been corrected.

c) Please explain the “T” in the legend for Fig. 2a,d.

Answer: This has been corrected.

2) Methods description

a) On p. 13 line 3, the superfluous “D_u = 13.5” should be removed.

Answer: D_w is in fact equal to 12 times D_u. The line should read “... Du =Dv =1.125,Dw =12 · Du =13.5, ...”

This has now been corrected.

b) $z^{\text{skin_top}}$ on p.15 does not seem to be defined. Please clarify.

Answer: This has been corrected.

3) Simulations and software

a) It would be useful to give a rough idea to the readers how long the simulations take on an average computer, such that the speed-up benefit on p. 9 line 28 can be appreciated (e.g. does the 10-20 time gain correspond to 10 seconds, 10 hours, or 10 days?). This information can currently only be found in the software package, and the README file indicates that the gain will most likely be in the seconds to minutes range.

Answer: This information has now been incorporated into the text.

b) Please add the red dot at the origin in Fig. 5a, just like in Supplementary Movie 1.

Answer: This has been corrected.

c) To make the software widely applicable, it would be useful to add the other network types to the software as well (e.g. /gaussian_bumps/*.txt etc.) and to extend the README file with other input options for the simulations, so that 2D projection simulations etc. can also easily be executed.

Answer: The Gaussian bump scheme has been added in the code provided and the README file has been updated accordingly.

end -

Reviewer #1 (Remarks to the Author):

The authors have mostly addressed my concerns. Before the manuscript can be accepted for publication, however, I think that the authors should consider the following points. They mostly pertain to the presentation of the authors' arguments, rather than to the arguments themselves. For this reason, I do not want to insist that the authors make all of the changes that I have suggested, but I do believe that these suggestions could make the manuscript more readable. Apologies about their number (and especially the number of minor comments)!

(1) The first point I raised in my first report was the structure of the paper, with its long introduction summarising Manukyan et al. (2017), which I shall again refer to as M2017. I agree with the authors that the introduction needs to balance brevity and setting out the biological and physical questions clearly. I therefore appreciate the removal of the paragraph discussing the slowing down of the colour switching rate observed in M2017. I believe, however, that the introduction could be streamlined further by removing "More specifically [...] biological evolution" [lines 23-30 of page 4], which discusses results of M2017 which are, as far I can see, never referred to again, and not relevant to the problem that the authors discuss in this paper.

(2) The next point I raised was to do with the biological basis for the RD models used. I appreciate the new "Discussion" section in the revised manuscript, and I think that the discussion of the biological basis for the use of a RD is really helpful. I understand that the other referees raised the issue of "agent-based models" more explicitly, but I for one would feel that the authors would be justified in using the simpler RD models just because the complete patterning mechanism is not yet understood (even though some more components of the puzzle could perhaps be introduced into the AB models). Anyway, the authors' important contribution is, as they say, to show that RD models coupled to 3D geometric effects can explain the CA patterning. This is in some sense completely irrespective of the existence of AB modelling approaches (even though I agree that the existence of such approaches needs to be mentioned).

(a) For this reason, I was wondering whether the almost philosophical comparison of the different modelling approaches in the discussion is really necessary (or indeed appropriate in a research article, as opposed to a review). However, I certainly do not want to insist that this be removed (since it was put in because of the questions raised by the other referees). Even if this comparison is kept in the paper, I feel that the discussion of the biological basis for an RD patterning model (currently the last paragraph of the discussion) would need to come before the paragraph extolling the virtues of Turing models: no matter how mathematically elegant they may be, if there were no biological basis for the use of a RD model, they should not be used for this problem!

(b) A long paragraph in the discussion deals with the subspace of the skin actually occupied by pigment cells, partly because of a question raised by Referee 3. I do not want to denigrate the authors' experimental achievements in acquiring realistic 3D geometries, but I am still wondering: how much do we really learn from the computations in realistic geometries given that we do not know what the correct RD model is and any RD model used is therefore a possibly very rough approximation? This is especially relevant, because, as I noted in my first report, even if we accept that the geometric effects cause CA behaviour in all RD systems (for sufficient thinning), we do not know what thinning threshold is in the unknown RD system describing the patterning. (It might be good to mention the evolutionary distance between fish and lizards given in the authors' reply in the paper, too, when explaining the approximations involved in using the RD model of Nakamasu et al.

to describe these patterning processes that are still only understood approximately.) I therefore feel that the limitations of the model would swamp these questions of precise domain extent. Again, I do not want to belittle the importance of the results in the paper, and I do agree with the comment of Referee 3 which the authors quote in their reply to my first report, that the hypothesis of M2017 needs testing, but my point was the following: assume that the authors had worked on this problem some years ago, when the 3D GPU computations were impossible and HREM microscopy was not available. They could still have written an almost equivalent paper, by saying that the scales thin at their boundary, that the 3D diffusion problem in a thin domain is asymptotically described by the Fick-Jacobs equation, which they could have solved in 2D to reveal the CA behaviour.

(3) The final points in my first report pertained to the dimensionality reduction scheme. While I agree with the authors' response, I think they should clarify some of their edits to the paper:

- A reference should probably be added to "Following standard procedures". Then, the justification paragraph should appear after "in the z direction" ("This is justified by [...]"). I would also encourage the authors to not just state the ingredients for the justification, but either (a) explain how they allow one to neglect the z-variation (as suggested in my first report) or (b) add a reference.

- It does not make sense to talk about "large enough values of $D(x)$ " or "low $D(x)$ ", since D is a dimensional quantity. The approximation requires $\partial z_0/\partial x$ and $\partial z/\partial x$ to be small, which I suppose are also the conditions under which Bradley's result holds. This should be clarified.

Minor comments:

(1) The word "dynamics", with which the authors have replaced unidiomatic uses of "dynamic", is plural. The authors should therefore say e.g. "CA dynamics" and not "a CA dynamics" [abstract], "these dynamics gradually generate" and not "this dynamics gradually generates" [page 4, line 17], etc.

(2) In the abstract, "We previous identified" should of course be "We previously identified"; also, the use of "identify" is unidiomatic, and the second sentence then suddenly mentions diffusion constants before mentioning the corresponding RD system more than a line later. I for one would find the following much more readable:

"We previously showed that the adult skin colour pattern of ocellated lizards can be understood as a probabilistic cellular automaton (CA) of skin scales. We additionally suggested that this discrete CA is produced by a continuous two-dimensional (2D) reaction-diffusion (RD) system of colour pattern development because of a reduction of the diffusion coefficients at the borders of the scales, which results in turn from the thinning of the skin there. Here, we use [...] geometries to demonstrate that skin thickness variation on its own is sufficient to cause scale-by-scale coloration and CA dynamics during [...] of the labyrinthine colour pattern of the lizard skin."

(3) The phrase "cells-as-agents models" is only used once [page 4, line 6]. This should probably be replaced with "agent-based models".

(4) I appreciate that the authors now use the present tense consistently in the "Results" section. I would suggest that they say "indicated" in line 20 on page 4 to use the past tense consistently in the introduction, when describing the results of M2017. Similarly, I would find using the present perfect

("Here, we have tested" rather than "Here, we test") more natural in the conclusion.

(5) I understand that the sentence "Note that, for all these RD systems [...] elements." [lines 5-9 on page 8] was introduced in response to a question of Referee 3, but it may not be clear to the reader why they should want to compare the steady-state pattern lengthscale to the size of the hexagons, so some explanation should be added.

(6) Here are some more typos or unidiomatic phrases:

- [page 5, line 21] "on the GPU" -> "on a GPU";

- [page 5, lines 25-27] "making the system [...] of scales" -> "solution of the system of RD equations on the reconstructed geometry of the skin scales produces a spatially discrete pattern exactly superposed on the scale lattice";

- [page 6, line 3] "necessary and sufficient" -> "sufficient" (the argument does not prove that scale thinning is the only mechanism that can give CA behaviour);

- [page 7, e.g. line 6] "test if" -> "test whether";

- [page 9, e.g. line 17] "the Fick-Jacobs' and Bradley [...] methods" -> "the Fick-Jacobs and Bradley methods";

- [page 9, line 22] "the RD components' mean concentrations" -> "the mean concentrations of the RD components" (such constructions, also used in the caption of Fig. 2, are unidiomatic);

- [page 10, line 32] "decrease" -> "decreases";

- [page 11, line 14] "circumstance" -> "possibility"; I do not understand the last sentence of this paragraph;

- [page 12, line 7] "skin labyrinthine pattern" -> "labyrinthine skin pattern";

- [page 15, line 7] "If one considers" -> "If one makes";

- [page 16, line 9] "using RD models vastly different from [...] vertebrates" -> e.g. "using RD models vastly different from that of Nakamasu et al." since the RD used is not the (unknown) model describing the patterning;

- [page 18, line 13] "by using another value than" -> "by using a value other than";

- [page 18, line 16] "node" -> "nodes";

- [page 18, line 17] "Euler forward method" -> "the Euler forward method";

- [page 19, line 3] "assignment" -> "assignments";

- [page 19, line 35] "an euthanised" -> "a euthanised";

- [page 23] the definition of z_0 ("where $z_0(x) = [\dots]$ ") should appear after the first displayed equation (" $D(x) = [\dots]$ ") giving Bradley's result;

- [page 25, second paragraph] "Average curvature" -> "The average curvature"; "Menger curvature" -> "the Menger curvature" (reference?); "To get [...] resolution" -> "To scale out image resolution effects";

- [page 25, third paragraph] "Prism [...] error" -> "The prism [...] error"; "where" -> "in which".

- [caption of Fig. 1] "at 2 [...] ages" -> "at the age of 104 weeks (D), but get obliterated in older lizards";

- [e.g. caption of Figs. 2,3] "graphs show" can be removed.

Reviewer #3 (Remarks to the Author):

In the revised manuscript, the authors have appropriately addressed all reviewers' comments and included additional data and analyses to bolster their conclusions.

ANSWERS TO REVIEWER COMMENTS

Reviewer #1 (Remarks to the Author):

The authors have mostly addressed my concerns. Before the manuscript can be accepted for publication, however, I think that the authors should consider the following points. They mostly pertain to the presentation of the authors' arguments, rather than to the arguments themselves. For this reason, I do not want to insist that the authors make all of the changes that I have suggested, but I do believe that these suggestions could make the manuscript more readable. Apologies about their number (and especially the number of minor comments)!

Answer: we thank the Reviewer for her/his constructive and extensive review. The new comments are well taken and useful to produce a better final manuscript.

(1) The first point I raised in my first report was the structure of the paper, with its long introduction summarising Manukyan et al. (2017), which I shall again refer to as M2017. I agree with the authors that the introduction needs to balance brevity and setting out the biological and physical questions clearly. I therefore appreciate the removal of the paragraph discussing the slowing down of the colour switching rate observed in M2017. I believe, however, that the introduction could be streamlined further by removing "More specifically [...] biological evolution" [lines 23-30 of page 4], which discusses results of M2017 which are, as far I can see, never referred to again, and not relevant to the problem that the authors discuss in this paper.

Answer: we streamlined the paragraph as requested, keeping only the reference to a CA emerging from the evolutionary process; a very important concept indeed, especially for members of the community of evolutionary biologists, who are likely to be interested by our results.

(2) The next point I raised was to do with the biological basis for the RD models used. I appreciate the new "Discussion" section in the revised manuscript, and I think that the discussion of the biological basis for the use of a RD is really helpful. I understand that the other referees raised the issue of "agent-based models" more explicitly, but I for one would feel that the authors would be justified in using the simpler RD models just because the complete patterning mechanism is not yet understood (even though some more components of the puzzle could perhaps be introduced into the AB models). Anyway, the authors' important contribution is, as they say, to show that RD models coupled to 3D geometric effects can explain the CA patterning. This is in some sense completely irrespective of the existence of AB modelling approaches (even though I agree that the existence of such approaches needs to be mentioned).

Answer: we fully agree with the reviewer and added a very short note along these lines (RD models justified by the complete patterning mechanism not being fully understood) in the discussion section.

(a) For this reason, I was wondering whether the almost philosophical comparison of the different modelling approaches in the discussion is really necessary (or indeed appropriate in a research article, as opposed to a review). However, I certainly do not want to insist that this be removed (since it was put in because of the questions raised by the other referees). Even if this comparison is kept in the paper, I feel that the discussion of the

biological basis for an RD patterning model (currently the last paragraph of the discussion) would need to come before the paragraph extolling the virtues of Turing models: no matter how mathematically elegant they may be, if there were no biological basis for the use of a RD model, they should not be used for this problem!

Answer: we agree with the reviewer and moved the paragraph accordingly.

(b) A long paragraph in the discussion deals with the subspace of the skin actually occupied by pigment cells, partly because of a question raised by Referee 3. I do not want to denigrate the authors' experimental achievements in acquiring realistic 3D geometries, but I am still wondering: how much do we really learn from the computations in realistic geometries given that we do not know what the correct RD model is and any RD model used is therefore a possibly very rough approximation? This is especially relevant, because, as I noted in my first report, even if we accept that the geometric effects cause CA behaviour in all RD systems (for sufficient thinning), we do not know what thinning threshold is in the unknown RD system describing the patterning. (It might be good to mention the evolutionary distance between fish and lizards given in the authors' reply in the paper, too, when explaining the approximations involved in using the RD model of Nakamasu et al. to describe these patterning processes that are still only understood approximately.) I therefore feel that the limitations of the model would swamp these questions of precise domain extent. Again, I do not want to belittle the importance of the results in the paper, and I do agree with the comment of Referee 3 which the authors quote in their reply to my first report, that the hypothesis of M2017 needs testing, but my point was the following: assume that the authors had worked on this problem some years ago, when the 3D GPU computations were impossible and HREM microscopy was not available. They could still have written an almost equivalent paper, by saying that the scales thin at their boundary, that the 3D diffusion problem in a thin domain is asymptotically described by the Fick-Jacobs equation, which they could have solved in 2D to reveal the CA behaviour.

Answer: Here, we strongly prefer to keep this paragraph motivated by a question from Reviewer 3. In a sense, it is a discussion similar to the one pertaining to RD vs AB. Indeed, RD models might be interpreted as approximations of more complex AB models, but hexagonal prisms can also be interpreted as an approximation of the real skin geometry, and the skin thickness variation can be interpreted as an approximation of the spatial distribution of the actual 3D field in which the cell interactions occur. These questions have been raised in multiple occasions since the publication of M2017. We therefore think that the question of Reviewer 3 is very relevant and discussing it is of interest to the community.

(3) The final points in my first report pertained to the dimensionality reduction scheme. While I agree with the authors' response, I think they should clarify some of their edits to the paper:

- A reference should probably be added to "Following standard procedures". Then, the justification paragraph should appear after "in the z direction" ("This is justified by [...]"). I would also encourage the authors to not just state the ingredients for the justification, but either (a) explain how they allow one to neglect the z-variation (as suggested in my first report) or (b) add a reference.

Answer: We modified the text to now fully justify neglecting the z-variation of c.

- It does not make sense to talk about "large enough values of $D(x)$ " or "low $D(x)$ ", since D is a dimensional quantity. The approximation requires $\partial z_0/\partial x$ and $\partial z/\partial x$ to be small, which I

suppose are also the conditions under which Bradley's result holds. This should be clarified.

Answer: The Reviewer is right. We have now clarified the statement.

Minor comments:

(1) The word "dynamics", with which the authors have replaced unidiomatic uses of "dynamic", is plural. The authors should therefore say e.g. "CA dynamics" and not "a CA dynamics" [abstract], "these dynamics gradually generate" and not "this dynamics gradually generates" [page 4, line 17], etc.

Answer: This change is now incorporated across the manuscript.

(2) In the abstract, "We previous identified" should of course be "We previously identified"; also, the use of "identify" is unidiomatic, and the second sentence then suddenly mentions diffusion constants before mentioning the corresponding RD system more than a line later. I for one would find the following much more readable:

"We previously showed that the adult skin colour pattern of ocellated lizards can be understood as a probabilistic cellular automaton (CA) of skin scales. We additionally suggested that this discrete CA is produced by a continuous two-dimensional (2D) reaction-diffusion (RD) system of colour pattern development because of a reduction of the diffusion coefficients at the borders of the scales, which results in turn from the thinning of the skin there. Here, we use [...] geometries to demonstrate that skin thickness variation on its own is sufficient to cause scale-by-scale coloration and CA dynamics during [...] of the labyrinthine colour pattern of the lizard skin."

Answer: We appreciate the suggestions of the Reviewer but, if these were all taken into account, the abstract would substantially exceed the number of words allowed by the journal. Hence, we introduced as many of the suggested changes, as well as others along the lines proposed by the Reviewer, while not exceeding the 150 words limit.

(3) The phrase "cells-as-agents models" is only used once [page 4, line 6]. This should probably be replaced with "agent-based models".

Answer: We now write at the end of page 3: "Note that, 'cells-as-agents' approaches²²⁻²⁴ (referred to as 'agent-based' (AB) below), some of them²³ incorporating ..."
and we then use 'AB' throughout the manuscript.

(4) I appreciate that the authors now use the present tense consistently in the "Results" section. I would suggest that they say "indicated" in line 20 on page 4 to use the past tense consistently in the introduction, when describing the results of M2017. Similarly, I would find using the present perfect ("Here, we have tested" rather than "Here, we test") more natural in the conclusion.

Answer: These changes have been introduced.

(5) I understand that the sentence "Note that, for all these RD systems [...] elements." [lines 5-9 on page 8] was introduced in response to a question of Referee 3, but it may not be clear to the reader why they should want to compare the steady-state pattern lengthscale to the size of the hexagons, so some explanation should be added.

Answer: The explanation has been introduced in the manuscript.

(6) Here are some more typos or unidiomatic phrases:

- [page 5, line 21] "on the GPU" -> "on a GPU";

Answer: This has been corrected.

- [page 5, lines 25-27] "making the system [...] of scales" -> "solution of the system of RD equations on the reconstructed geometry of the skin scales produces a spatially discrete pattern exactly superposed on the scale lattice";

Answer: This has been corrected.

- [page 6, line 3] "necessary and sufficient" -> "sufficient" (the argument does not prove that scale thinning is the only mechanism that can give CA behaviour);

Answer: This has been corrected.

- [page 7, e.g. line 6] "test if" -> "test whether";

Answer: This has been corrected.

- [page 9, e.g. line 17] "the Fick-Jacobs' and Bradley [...] methods" -> "the Fick-Jacobs and Bradley methods";

Answer: This has been corrected throughout.

- [page 9, line 22] "the RD components' mean concentrations" -> "the mean concentrations of the RD components" (such constructions, also used in the caption of Fig. 2, are unidiomatic);

Answer: These have been corrected.

- [page 10, line 32] "decrease" -> "decreases";

Answer: This has been corrected.

- [page 11, line 14] "circumstance" -> "possibility"; I do not understand the last sentence of this paragraph;

Answer: The change has been introduced and the last sentence has been modified to clarify the point.

- [page 12, line 7] "skin labyrinthine pattern" -> "labyrinthine skin pattern";

Answer: This has been corrected.

- [page 15, line 7] "If one considers" -> "If one makes";

Answer: This has been corrected.

- [page 16, line 9] "using RD models vastly different from [...] vertebrates" -> e.g. "using RD models vastly different from that of Nakamasu et al." since the RD used is not the (unknown) model describing the patterning;

Answer: This has been corrected.

- [page 18, line 13] "by using another value than" -> "by using a value other than";

Answer: This has been corrected.

- [page 18, line 16] "node" -> "nodes";

Answer: This has been corrected.

- [page 18, line 17] "Euler forward method" -> "the Euler forward method";
Answer: This has been corrected.
- [page 19, line 3] "assignment" -> "assignments";
Answer: This has been corrected.
- [page 19, line 35] "an euthanised" -> "a euthanised";
Answer: This has been corrected.
- [page 23] the definition of z_0 ("where $z_0(x) = [\dots]$ ") should appear after the first displayed equation (" $D(x) = [\dots]$ ") giving Bradley's result;
Answer: This has been corrected.
- [page 25, second paragraph] "Average curvature" -> "The average curvature"; "Menger curvature" -> "the Menger curvature" (reference?); "To get [...] resolution" -> "To scale out image resolution effects";
Answer: These have been corrected.
- [page 25, third paragraph] "Prism [...] error" -> "The prism [...] error"; "where" -> "in which".
Answer: These have been corrected.
- [caption of Fig. 1] "at 2 [...] ages" -> "at the age of 104 weeks (D), but get obliterated in older lizards";
Answer: These have been corrected.
- [e.g. caption of Figs. 2,3] "graphs show" can be removed..
Answer: These have been corrected in Figs 2 and 4.

Reviewer #3 (Remarks to the Author):

In the revised manuscript, the authors have appropriately addressed all reviewers' comments and included additional data and analyses to bolster their conclusions.

Answer: we thank the Reviewer for her/his review and we are glad that he/she finds our revised version appropriate.

end -

Reviewer #1 (Remarks to the Author):

I thank the authors for making a commendable effort to clarify the presentation of their arguments further. I support publication of the revised manuscript.

In my previous report, I did not intend to appear to suggest that the authors should remove the paragraph discussing possible limitations of their analysis due to their estimates of the subspace of the skin actually occupied by pigment cells - apologies! Indeed, I fully agree that this is an important question. However, what I wanted to point out is that it is not clear to me (1) how these limitations compare to the limitations that result from lack of knowledge of the precise molecular basis for the patterning and the consequent use of generic RD models and therefore (2) how much the computations in a realistic geometry (but with a generic RD model) tell us about the actual biological system. While I therefore still think that this is an important question, I can also see that this problem is somewhat distinct from the geometric problem that the authors address in this paper, and I therefore still support publication.

One final minor comment:

I was not aware of the word length limit for the abstract, but could I still suggest using the words "reaction diffusion (RD) process" before "diffusion coefficients" if possible, since the reader may wonder otherwise why diffusion coefficients enter the problem? One possible phrasing might be:

"We additionally suggested that the canonical continuous 2D reaction-diffusion (RD) process of colour pattern development is transformed into this discrete CA by reduced diffusion coefficients at the borders of scales (justified by the corresponding thinning of the skin)."